# Reparameterization Proximal Policy Optimization

**Hai Zhong** [1]  **Xun Wang** [1]  **Zhuoran Li** [1]  **Longbo Huang** [1]

## Abstract

By leveraging differentiable dynamics, Reparameterization Policy Gradient (RPG) achieves high sample efficiency. However, current approaches are hindered by two critical limitations: the under-utilization of computationally expensive dynamics Jacobians and inherent training instability. While sample reuse offers a remedy for under-utilization, no prior principled framework exists, and naive attempts risk exacerbating instability. To address these challenges, we propose Reparameterization Proximal Policy Optimization (RPO). We first establish that under sample reuse, RPG naturally optimizes a PPO-style surrogate objective via Backpropagation Through Time, providing a unified framework for both on- and off-policy updates. To further ensure stability, RPO integrates a clipped policy gradient mechanism tailored for RPG and employs explicit Kullback-Leibler divergence regularization. Experimental results demonstrate that RPO maintains superior sample efficiency and consistently outperforms or achieves state-of-the-art performance across diverse tasks.

## 1. Introduction

Reparameterization Policy Gradient (RPG) (Mohamed et al., 2020; Amos et al., 2021) is a policy gradient method that computes the policy gradient using the reparameterization trick (Kingma & Welling, 2014; Rezende et al., 2014). Unlike REINFORCE (Williams, 1992; Sutton et al., 1999), RPG directly backpropagates through the trajectory to obtain a policy gradient estimate. This approach has become increasingly attractive with the recent rise of differentiable simulators (Hu et al., 2020; Xu et al., 2021; Xing et al., 2025; You et al., 2025). The applicability of RPG methods extends across a broad spectrum of robotic domains, including autonomous driving, quadrupedal locomotion, and agile flight (Song et al., 2024; Nachkov et al., 2026; Zhang et al., 2024). Notably, the exceptional sample efficiency of RPG has driven breakthroughs in training policies directly in physical environments, as exemplified by recent achievements in real-world quadrotor flight (Pan et al., 2025).

However, existing RPG-based approaches face two primary challenges. First, backpropagating through system dynamics is computationally expensive due to the calculation of dynamics Jacobians. Yet, prior on-policy methods lack a sample reuse mechanism (more specifically, reusing dynamics Jacobians): they discard costly dynamics Jacobians after a single policy update. Reusing these computationally expensive dynamics Jacobians could further boost RPG's sample efficiency and shorten the wall-clock training time of RPG-based approaches. Sample reuse is particularly desirable for learning RL policies in the real world with RPG, where data collection is time-consuming and fully utilizing each sample is paramount.

Second, RPG is notoriously prone to optimization instability, often suffering from exploding or vanishing gradients in environments with non-smooth dynamics or long horizons (Metz et al., 2021; Suh et al., 2022). Even with state-of-the-art (SOTA) variance reduction techniques, such as short-horizon rollouts in SHAC (Xu et al., 2021) and entropy regularization in SAPO (Xing et al., 2025), we empirically observe that RPG suffers from training instability, as shown in Figure 1. This creates a dilemma: while sample reuse is desirable for efficiency, it inherently exacerbates optimization instability since it increases the update-to-data ratio, necessitating an explicit mechanism to constrain policy updates.

In this work, we propose **Reparameterization Proximal Policy Optimization (RPO)** to address these limitations. First, we establish a principled sample reuse mechanism for RPG. This is achieved by demonstrating that, under sample reuse, RPG naturally aligns with a PPO-like surrogate objective via backpropagation through time (BPTT), providing a unified framework for both on- and off-policy updates. This formulation enables RPO to achieve high sample efficiency. Second, to maintain stability even with sample reuse, RPO incorporates a clipped policy gradient mechanism tailored

[1] Institute for Interdisciplinary Information Sciences (IIIS), Tsinghua University. Correspondence to: Longbo Huang <longbohuang@tsinghua.edu.cn>.

*Proceedings of the 43rd International Conference on Machine Learning*, Seoul, South Korea. PMLR 306, 2026. Copyright 2026 by the author(s).

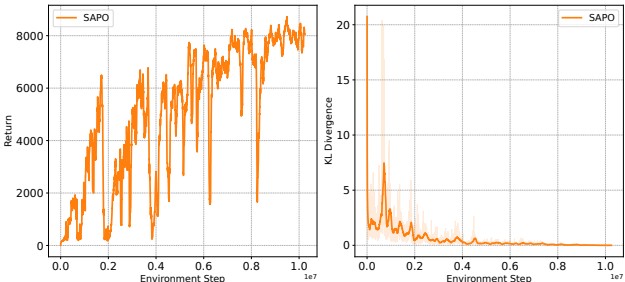

*Figure 1.* An example of SAPO's training instability in the Humanoid task. Large KL divergence (both smoothed and raw curves shown on the right) correspond to sudden performance drops.

to the specific characteristics of RPG, which constrains updates driven by large importance weights. Furthermore, we enhance stability via an explicit KL divergence regularization term, as we empirically observe that clipping alone is insufficient. Finally, RPO remains fully compatible with and benefits from existing variance reduction methods for RPG.

We conduct experiments on a suite of locomotion and manipulation tasks using two differentiable simulators, DFlex (Xu et al., 2021; Georgiev et al., 2024) and Rewarped (Xing et al., 2025). Experimental results show that RPO achieves superior sample efficiency and consistently achieves state-of-the-art performances across all tasks. We believe RPO's sample reuse mechanism holds great promise for future RPG-based applications in learning real-world RL policies, as utilizing each rollout trajectory to its full potential is clearly critical.

To summarize, our main contributions are: **(i)** We show that with the sample reuse mechanism, RPG is naturally linked to a PPO-like surrogate objective, providing a unified framework for both on- and off-policy updates. **(ii)** We propose **Reparameterization Proximal Policy Optimization**, which achieves stable and sample-efficient learning by synergizing sample reuse with a tailored importance weight clipping mechanism and explicit KL regularization. **(iii)** We conduct extensive experiments on a suite of locomotion and manipulation tasks using the differentiable simulators DFlex and Rewarped, demonstrating RPO's high sample efficiency and strong performance.

## 2. Related Work

**Policy Gradient Estimators.** One classical class of policy gradient estimators is based on the score function, such as the REINFORCE gradient estimator (Williams, 1992; Sutton et al., 1999). Many policy gradient methods, such as PPO and TRPO (Schulman et al., 2017; 2015), rely on variants of the REINFORCE gradient estimator. One limitation

of the REINFORCE gradient is its high variance, which results in low sample efficiency.

On the other hand, if one has access to the underlying dynamic model, either through differentiable simulators (Xu et al., 2021; Hu et al., 2020; Xing et al., 2025) or learned world models (Hafner et al., 2020; Amos et al., 2021), another type of policy gradient named Reparameterization Policy Gradient, which is based on the reparameterization trick (Kingma & Welling, 2014; Rezende et al., 2014), can be obtained. Using the reparameterization trick (Kingma & Welling, 2014; Rezende et al., 2014), RPG directly backpropagates through the trajectory and obtains an unbiased estimate of the policy gradient. By contrast, the REINFORCE gradient estimator does not need to backpropagate through the entire computational graph and only relies on local computation (Parmas, 2018). Since RPG utilizes the gradients of the dynamics model, RPG typically enjoys less variance than the REINFORCE gradient estimator (Mohamed et al., 2020).

**RPG-based Reinforcement Learning Algorithms.** It is well known that RPG obtained by vanilla backpropagation through time over a long time horizon suffers from the vanishing/exploding gradient problem (Suh et al., 2022; Metz et al., 2021; Zhang et al., 2023b; Ma et al., 2024). This phenomenon is amplified when dealing with stiff dynamics, such as contact (Zhang et al., 2023a; Suh et al., 2022; Zhong et al., 2023; Pang et al., 2023). RPG can exhibit a large variance when the gradient magnitude is large, which renders the underlying reinforcement learning algorithm unstable, struggling with non-convex loss landscapes.

Several works (Parmas et al., 2018; 2023; Suh et al., 2022) weight and combine RPG and REINFORCE according to their variance, while AGPO (Gao et al., 2024) further combines RPG with gradients of Q-functions. SHAC and AHAC (Xu et al., 2021; Georgiev et al., 2024) reduce the variance of RPG by only backpropagating through a truncated length of the trajectory, aided by a value function to estimate future returns. MB-MIX (Zhang et al., 2025) backpropagates a mixture of trajectories with different lengths to better balance the bias-variance trade-off. GI-PPO (Son et al., 2023) first optimizes the policy using RPG, then uses the REINFORCE gradient to perform further off-policy updates in the PPO style. However, the gradients computed by REINFORCE are not only of lower quality than those from RPG, but can also conflict with each other, thereby degrading sample efficiency and performance. Entropy is also introduced to regularize RPG-based policy updates and promote exploration (Xing et al., 2025; Amos et al., 2021).

While these works improve various aspects of RPG, the question of how to enable stable sample reuse for RPG remains largely unaddressed, motivating our approach.

## 3. Preliminaries

### 3.1. Reinforcement Learning Formulation

In this work, we consider problems formulated as a Markov Decision Process (MDP) (Sutton & Barto, 2018). An MDP is formally defined by a tuple $(\mathcal{S}, \mathcal{A}, p, r, p_0, \gamma)$, where $\mathcal{S}$ is the set of states, $\mathcal{A}$ is the set of actions, $p : \mathcal{S} \times \mathcal{A} \times \mathcal{S} \to [0, 1]$ is the state transition probability function, $r : \mathcal{S} \times \mathcal{A} \to \mathbb{R}$ is the reward function, $s_0$ is the initial state, $p_0(s_0)$ is the initial state distribution, and $\gamma \in [0, 1)$ is the discount factor.

The goal of reinforcement learning (RL) is to find the optimal parameter $\theta^*$ for a parameterized stochastic policy $\pi_\theta$. A parameterized stochastic policy $\pi_\theta(a|s)$ specifies the probability distribution over actions $a \in \mathcal{A}$ given a state $s \in \mathcal{S}$. The optimal parameter $\theta^*$ maximizes the expected discounted cumulative reward:

$$\theta^* = \arg\max_\theta J(\theta) = \arg\max_\theta \mathbb{E}_{\tau \sim \pi_\theta}[R(\tau)] \quad (1)$$

where $\tau = (s_0, a_0, s_1, a_1, \dots)$ is the trajectory generated by following the policy $\pi_\theta$. Here, $r(s_t, a_t)$ denotes the reward received at time step $t$, and $R(\tau) = \sum_{t=0}^{\infty} \gamma^t r(s_t, a_t)$ is the discounted cumulative reward for the trajectory $\tau$.

### 3.2. Reparameterization Policy Gradient

RPG uses the reparameterization trick (Kingma & Welling, 2014; Rezende et al., 2014; Haarnoja et al., 2018) to sample an action $a_t$ from a Gaussian policy $\pi_\theta(a_t|s_t)$. Specifically, the policy network first predicts the mean $\mu_\theta(s_t)$ and standard deviation $\sigma_\theta(s_t)$, and then combines them with a reparameterization noise $\epsilon_t$ at time step $t$:

$$a_t = \mu_\theta(s_t) + \sigma_\theta(s_t) \cdot \epsilon_t, \quad \text{where } \epsilon_t \sim \mathcal{N}(0, \mathcal{I}). \quad (2)$$

Here, $\sigma_\theta(s_t)$ is a diagonal matrix with positive elements on the diagonal (this suggests that each action dimension is independent, which is a common practice in reparameterization Gaussian policies). We denote this transformation as $a_t = f_\theta(\epsilon_t; s_t)$.

We focus on deterministic system dynamics $s_{t+1} = g(s_t, a_t)$. Consistent with prior works (Xu et al., 2021; Georgiev et al., 2024; Xing et al., 2025), we adopt the following assumption regarding the dynamics and rewards.

**Assumption 1.** The system dynamics $g(s, a)$ and the reward function $r(s, a)$ are differentiable w.r.t. state $s$ and action $a$.

With the reparameterization trick, RPG can backpropagate through dynamics by computing Jacobians, $\frac{\partial s_{t+1}}{\partial a_t}$ and $\frac{\partial s_{t+1}}{\partial s_t}$, to obtain a policy gradient estimate:

$$\nabla_\theta J(\theta) = \mathbb{E}_{s_0, \epsilon_0, \epsilon_1, \dots}[\nabla_\theta R(\tau)]. \quad (3)$$

Note that the expectation is taken with respect to the initial state distribution and sampled noise at different time steps.

However, full backpropagation through long trajectories could lead to high gradient variance and unstable training. Short-Horizon Actor-Critic (SHAC) (Xu et al., 2021), addresses this by backpropagating through only a short horizon of the trajectory, using a value function to capture the long-term return. SHAC's variant of RPG is as follows:

$$\nabla_\theta[R(\tau_{t_0:t_0+h-1}) + \gamma^h V(s_{t_0+h})], \quad (4)$$

where $t_0$ is the starting time step for the trajectory, $h$ is the short horizon, $R(\tau_{t_0:t_0+h-1})$ is the cumulative reward of the trajectory within the short horizon, and $V(s_{t_0+h})$ is the value function's estimate of the future return. In practice, SHAC accumulates the gradient in Eq. (4) over a batch of short-horizon segments and normalizes by the product of the batch size and the rollout horizon $h$.

### 3.3. Surrogate Objective

PPO (Schulman et al., 2017) and TRPO (Schulman et al., 2015) optimize variants of the following surrogate objective function (Kakade, 2001):

$$L_{\pi_{\theta_{\text{old}}}}(\theta) = \int_s \sum_{t=0}^{\infty} \gamma^t p(s_t = s|\pi_{\theta_{\text{old}}}) \int_a A^{\pi_{\theta_{\text{old}}}}(s, a) \pi_\theta(a|s) \mathrm{d}a\mathrm{d}s \quad (5)$$

where $\pi_{\theta_{\text{old}}}$ is the behavior policy used to collect samples; $\sum_{t=0}^{\infty} \gamma^t p(s_t = s|\pi_{\theta_{\text{old}}})$ is the unnormalized state probability density induced by $\pi_{\theta_{\text{old}}}$; and $A^{\pi_{\theta_{\text{old}}}}(s, a)$ is the advantage function corresponding to $\pi_{\theta_{\text{old}}}$. This objective measures the performance of the new policy $\pi_\theta$, using the state distribution and advantages for the behavior policy $\pi_{\theta_{\text{old}}}$. PPO optimizes a clipped variant of this objective, while TRPO optimizes an explicit KL-constrained variant (Schulman et al., 2017; 2015).

By the law of the unconscious statistician (Grimmett & Stirzaker, 2001), we can rewrite the surrogate objective in the reparameterization form:

$$L_{\pi_{\theta_{\text{old}}}}(\theta) = \int_s d^{\pi_{\theta_{\text{old}}}}(s) \int_\epsilon A^{\pi_{\theta_{\text{old}}}}(s, a)|_{a=f_\theta(\epsilon;s)} p_{\text{std}}(\epsilon) \mathrm{d}\epsilon\mathrm{d}s \quad (6)$$

where $d^{\pi_{\theta_{\text{old}}}}(s)$ denotes the unnormalized state density induced by $\pi_{\theta_{\text{old}}}$ and $p_{\text{std}}(\epsilon)$ is the probability density function for standard Gaussian distribution. This surrogate objective differs fundamentally from the objective of the stochastic value gradient (SVG) (Heess et al., 2015; Amos et al., 2021), as (6) measures performance using the value function of the behavior policy, while SVG uses that of the current policy.

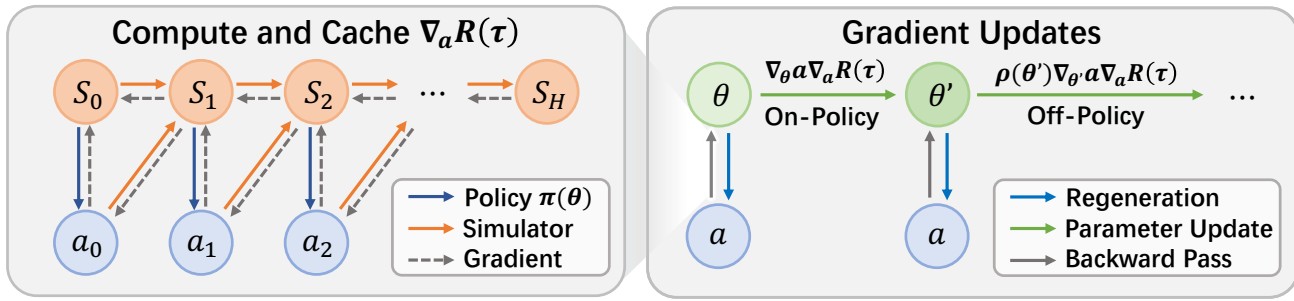

*Figure 2.* Computing the reparameterization policy gradient of the surrogate objective involves three steps: **(a)** Action-gradients are computed from rollouts via a single backward pass and cached. **(b)** These gradients are used directly for the initial, on-policy update. **(c)** For subsequent off-policy updates, the cached action-gradients are importance-weighted by $\rho(\theta')$ and reused, enabling stable sample reuse.

# 4. Reparameterization Proximal Policy Optimization

In this section, we introduce our proposed method: Reparameterization Proximal Policy Optimization. First, we show that with sample reuse (i.e., reusing computed action-gradients obtained via backpropagation through dynamics), RPG is indeed calculating the reparameterization policy gradient for the surrogate objective (Equation (6)). This provides a unified framework for both on-policy and off-policy updates for RPG. Based on this insight, we propose RPO to achieve stable sample reuse. RPO incorporates three key mechanisms: (i) optimizing the PPO-like surrogate objective via RPG, (ii) a policy gradient clipping mechanism designed for RPG, and (iii) an explicit KL regularization term. The overall algorithm is summarized in Algorithm 1.

## 4.1. Surrogate Objective for Policy Improvement

State-of-the-art RPG methods (Xu et al., 2021; Xing et al., 2025), are limited to a single policy update per batch, under-utilizing expensive BPTT gradients. While GI-PPO (Son et al., 2023) attempts sample reuse, it relies on REINFORCE gradients for off-policy updates, failing to utilize dynamics Jacobians. Its hybrid REINFORCE approach could introduce update conflicts and underutilize the low-variance RPG gradients.

A novel connection is established showing that sample reuse in RPG (i.e., reusing action-gradients computed via BPTT) is equivalent to calculating the reparameterization policy gradient for the surrogate objective, thereby providing a unified framework for both on-policy and off-policy updates.

Therefore, the first key component of RPO is introduced, the surrogate objective for policy improvement:

$$L_{\pi_{\theta_{\text{old}}}}(\theta) = \mathbb{E}_{s \sim d^{\pi_{\theta_{\text{old}}}}, \epsilon \sim p_{\text{std}}} \left[ A^{\pi_{\theta_{\text{old}}}}(s, f_\theta(\epsilon; s)) \right] \quad (7)$$

where $\pi_{\theta_{\text{old}}}$ is the policy before the update, $d^{\pi_{\theta_{\text{old}}}}(s)$ denotes the unnormalized state density induced by $\pi_{\theta_{\text{old}}}$, and $\epsilon$ is

sampled from the standard Gaussian distribution. We show that computing the reparameterization policy gradient for the surrogate objective is equivalent to reusing the action-gradient computed via BPTT, enabling sample reuse.

The reparameterization policy gradient of the surrogate objective with respect to policy parameter $\theta$ is as follows:

$$\nabla_\theta L_{\pi_{\theta_{\text{old}}}}(\theta)$$
$$= \int_s d^{\pi_{\theta_{\text{old}}}}(s) \int_\epsilon \left[ \nabla_\theta a \nabla_a A^{\pi_{\theta_{\text{old}}}}(s, a)|_{a=f_\theta(\epsilon;s)} p_{\text{std}}(\epsilon) \right] d\epsilon ds$$
$$= \int_s d^{\pi_{\theta_{\text{old}}}}(s) \int_\epsilon \left[ \nabla_\theta a \nabla_a Q^{\pi_{\theta_{\text{old}}}}(s, a)|_{a=f_\theta(\epsilon;s)} p_{\text{std}}(\epsilon) \right] d\epsilon ds.$$
$$(8)$$

Here $A^{\pi_{\theta_{\text{old}}}}(s, a) = Q^{\pi_{\theta_{\text{old}}}}(s, a) - V^{\pi_{\theta_{\text{old}}}}(s)$ and $V^{\pi_{\theta_{\text{old}}}}(s)$ is constant with respect to the parameter $\theta$ being optimized. Note that $\nabla_a A^{\pi_{\theta_{\text{old}}}}(s, a) = \nabla_a Q^{\pi_{\theta_{\text{old}}}}(s, a)$, since the value function $V^{\pi_{\theta_{\text{old}}}}(s)$ does not depend on actions.

The derivation in Equation (8) requires interchanging the order of differentiation and integration. We assume that standard regularity conditions for this interchange are satisfied. When these conditions do not hold, the expression no longer corresponds to a strict gradient of the surrogate objective; nevertheless, it remains a practical estimator that we find effective for optimizing the policy in practice. Empirically, irrespective of whether these regularity conditions are precisely satisfied, the resulting algorithm performs robustly across a range of differentiable simulation tasks, as we demonstrate in our experiments. This suggests that the proposed methods could work in practical settings.

### 4.1.1. DERIVING THE REPARAMETERIZATION GRADIENT VIA ACTION-GRADIENT REUSE

In this section, we show that computing the RPG for the surrogate objective naturally leads to reusing action-gradients for multiple policy updates (more details for this derivation are given in Appendix D).

As the first step, we collect a batch of rollouts with the behavior policy, setting $\pi_{\theta_{old}} = \pi_\theta$. To illustrate the connection, we consider the resulting infinite-horizon computational graph. We consider the gradients of the discounted cumulative return with respect to the action at each time step. From here, we clearly see that the action-gradient for time step $k$, $\nabla_{a_k} R(\tau) = \gamma^k \nabla_{a_k} \sum_{t=k}^{\infty} \gamma^{(t-k)} r(s_t, a_t)$, is exactly an unbiased Monte Carlo estimate of $\gamma^k \nabla_a Q^{\pi_{\theta_{old}}}(s_k, a_k)$, where $s_k$ and $a_k$ are sampled according to $p(s_k = s | \pi_{\theta_{old}})$ and $\pi_{\theta_{old}}(a_k | s_k)$. This holds because the reparameterization trick allows us to express $\gamma^k \nabla_a Q^{\pi_{\theta_{old}}}(s_k, a_k)$ as the expected gradient of the returns with respect to $a_k$ across all possible paths sampled via the reparameterization noise (as we assume deterministic dynamics). As shown in Figure 2, we cache these action-gradients for subsequent calculations (note that we only plot the trajectory for $h$ steps for illustration purposes). Note that this unbiased equality holds at the theoretical (infinite-horizon) level: if the cumulative return $R(\tau) = \sum_{t=0}^{\infty} \gamma^t r(s_t, a_t)$ could be evaluated exactly, $\nabla_{a_k} R(\tau)$ would be a strictly unbiased Monte Carlo estimate of $\gamma^k \nabla_a Q^{\pi_{\theta_{old}}}(s_k, a_k)$. In practice, the infinite-horizon sum cannot be computed directly; we approximate it using short-horizon rollouts with value-function bootstrapping, as described in Equation (4), which yields an accurate estimator of this theoretical target.

We first introduce the procedure for computing the off-policy RPG, as the on-policy gradient can be viewed as a special case of the off-policy gradient. Note that, in the first update epoch, the behavior policy $\pi_{\theta_{old}}$ is identical to the first epoch's policy $\pi_\theta$.

**Off-policy reparameterization policy gradient:** For off-policy updates, policy parameters are updated from $\theta$ to $\theta'$. The behavior policy $\pi_{\theta_{old}}$ (which generated the data) is now different from the current policy $\pi_{\theta'}$. Now we must compute the off-policy RPG gradient. As we will show, this naturally leads to reusing the action-gradients computed via BPTT.

The cached action-gradients $\nabla_{a_k} R(\tau)$ can be reused, but a computational path from the new policy parameters $\theta'$ to the action $a_k$ must be re-established to compute $\nabla_{\theta'} a_k$. This is achieved by regenerating the noise $\epsilon$ that is required for the current policy to produce $a_k$, which allows expressing the action as $a_k = f_{\theta'}(\epsilon; s_k)$, creating a new differentiable path.

As shown in Figure 2, the cached action-gradients are then backpropagated through this new path, yielding an estimate of $\gamma^k \nabla_{\theta'} a_k \nabla_{a_k} Q^{\pi_{\theta_{old}}}(s_k, a_k)|_{a_k = f_{\theta'}(\epsilon; s_k)}$ for each time step. We further weight each time step's gradient by the importance sampling ratio $\rho(\theta') = \frac{\pi_{\theta'}(a|s)}{\pi_{\theta_{old}}(a|s)}$ as an off-policy correction. Summing over time steps and averaging over the batch of trajectories yields a Monte Carlo estimate of the off-policy reparameterization policy gradient for the surrogate objective. A formal proof that this importance

ratio is the correct correction is given in Appendix D.

**On-policy reparameterization policy gradient:** In the first policy update epoch, the behavior policy $\pi_{\theta_{old}}$ is identical to the current policy $\pi_\theta$. The on-policy update follows the same procedure as off-policy updates, though for this first epoch, the importance weight ratio collapses to 1 (a special case of off-policy updates).

### 4.1.2. PRACTICAL IMPLEMENTATION FOR POLICY IMPROVEMENT

In this section, we provide the algorithmic details of the sample reuse mechanism for policy updates, building upon the previous derivations.

**Collecting rollouts and computing action-gradients.** Following SHAC (Xu et al., 2021), we collect a batch of $N$ short-horizon trajectories using the current policy $\pi_\theta$ as the behavior policy and bootstrap future returns via the value estimate of the terminal state. Note that if one wants to rigorously sample from $d^{\pi_{\theta_{old}}}(s)$ each time with a newly updated policy as the behavior policy, one should resample initial states from $p_0$ and start the rollout from there. However, in practice, we follow SHAC's procedure and continue from the previous rollout's truncated state, which works well empirically.

We then employ BPTT to compute and cache the corresponding action-gradients for each time step, denoted as $\nabla_a R(\tau)$, where $R(\tau)$ incorporates the terminal value estimate. Note that each trajectory in the batch is only back-propagated through once. As we previously discussed, the action-gradient for a specific time step $\nabla_{a_k} R(\tau)$ is an estimate of $\gamma^k \nabla_a Q^{\pi_{\theta_{old}}}(s_k, a_k)$. Since $A^{\pi_{\theta_{old}}}(s_k, a_k) = Q^{\pi_{\theta_{old}}}(s_k, a_k) - V^{\pi_{\theta_{old}}}(s_k)$ and $V^{\pi_{\theta_{old}}}(s_k)$ does not depend on the action $a_k$, $\nabla_{a_k} R(\tau)$ is similarly an estimate of $\gamma^k \nabla_a A^{\pi_{\theta_{old}}}(s_k, a_k)$.

**On-policy and Off-policy Updates.** We perform $M$ optimization epochs on a batch of cached action-gradients. The first update is on-policy, while all subsequent updates ($1 < m \le M$) are off-policy. Our method for computing the reparameterization policy gradient of the surrogate objective is unified across both cases. For each update step, we perform the following procedure.

First, to compute the gradient for the current policy $\pi_\theta$ using off-policy data, we must re-establish a computational path from policy network parameters $\theta$ to the actions. We achieve this by computing the noise $\epsilon$ that is required for the current policy to regenerate the actions stored in the rollout buffer:

$$\epsilon = f_\theta^{-1}(a; s), \tag{9}$$

where $f_\theta^{-1}(a; s)$ is the inverse of the reparameterization transform. With this recovered noise, we can express the action under the current policy as $a = f_\theta(\epsilon; s)$, which creates

a new computational graph connecting the current policy parameters $\theta$ to the action stored in the buffer. Note that SVG (Heess et al., 2015) has the same action regeneration mechanism.

Next, we introduce a novel **policy gradient clipping mechanism**, designed specifically for RPG. The proposed policy gradient clipping mechanism serves as a safeguard against numerical instability by filtering out samples with excessive importance weight ratios, thereby preventing action probabilities from becoming critically low.

Unlike PPO's clipping mechanism, our formulation clips the importance weight ratio asymmetrically and does not depend on the sign of the advantage function. This design is crucial because RPG, unlike REINFORCE, does not explicitly increase or decrease the log-likelihood of a specific action.

Specifically, let the importance weight ratio be $\rho(\theta) = \frac{\pi_\theta(a|s)}{\pi_{\theta_{\text{old}}}(a|s)}$. The gradient contribution from this action is non-zero only if $\rho(\theta)$ is within the clipping range, and is weighted by $\rho(\theta)$:

$$\begin{cases} \rho(\theta)\nabla_\theta a \nabla_a R(\tau), & \text{if } 1 - c_{low} \leq \rho(\theta) \leq 1 + c_{high}, \\ 0, & \text{otherwise}, \end{cases}$$
(10)

where $\nabla_a R(\tau)$ is the cached action-gradient. By performing this step for all actions in the buffer, we obtain the clipped policy gradient for the surrogate objective. Analogously to SHAC, this accumulated gradient is then normalized by the product of the batch size and the rollout horizon $h$.

### 4.2. Regularization Terms

While our importance ratio-based policy gradient clipping mechanism is designed for RPG (as detailed above), we empirically find that clipping alone is insufficient to fully ensure stability. More details are shown in ablation study and Appendix I. Hence, we incorporate a KL regularization (Kullback & Leibler, 1951) term, which penalizes large deviations from the behavior policy:

$$L_{KL}(\theta) = \mathbb{E}\left[D_{KL}(\pi_{\theta_{\text{old}}}(\cdot|s) \,||\, \pi_\theta(\cdot|s))\right]. \quad (11)$$

Note that this regularization only takes effect with sample reuse, as the KL divergence and its gradient are zero for the first on-policy update.

We also include an entropy bonus to encourage exploration (Haarnoja et al., 2018; Xing et al., 2025):

$$L_{ent}(\theta) = \mathbb{E}\left[H(\pi_\theta(\cdot|s))\right], \quad (12)$$

where $H(\pi_\theta(\cdot|s))$ denotes the entropy of $\pi_\theta$ at a given state.

---

**Algorithm 1** Reparameterization Proximal Policy Optimization (RPO)

---

1: Initialize policy parameters $\theta$ and value function parameters $\phi$.
2: **for** iteration $k = 1, 2, \ldots, K$ **do**
3:     Initialize empty buffer $\mathcal{B}$.
4:     Collect a batch of short-horizon trajectories by running policy $\pi_\theta$ in parallel environments and store them in buffer $\mathcal{B}$.
5:     **// Compute and cache action-gradients**
6:     Compute and cache the gradients of the discounted cumulative reward w.r.t. each action: $\nabla_a R(\tau)$.
7:     **for** policy update epochs $m = 1, 2, \ldots, M$ **do**
8:         Regenerate the actions stored in $\mathcal{B}$ (Equation (9)) with $\pi_\theta$.
9:         Backpropagate the clipped cached action-gradients to policy network parameters $\theta$, weighted by the importance weight ratios (Equation (10)).
10:        Compute the gradients of the KL divergence and entropy regularization terms (Equations (11),(12)).
11:        Combine the gradients and update the policy network parameters $\theta$.
12:     **end for**
13:     **for** value update epochs $l = 1, 2, \ldots, L$ **do**
14:         Update value function parameters $\phi$ by minimizing the regression loss (Equation (14)).
15:     **end for**
16: **end for**

---

### 4.3. Overall Policy Training Objective and Policy Update

The overall policy training objective is to maximize a weighted combination of all three components:

$$L_{policy}(\theta) = \lambda_{surr}L_{\pi_{\theta_{\text{old}}}}(\theta) - \lambda_{KL}L_{KL}(\theta) + \lambda_{ent}L_{ent}(\theta),$$
(13)

where $\lambda_{surr}, \lambda_{KL}$ and $\lambda_{ent}$ are the coefficients for the three terms. The three gradient components (from the surrogate objective and the two regularization terms) are combined according to their coefficients, and the final resulting gradient is used to update the policy parameters.

### 4.4. Value Function Training

The value function network is trained by minimizing the following regression loss (Xu et al., 2021):

$$L_\phi = \mathbb{E}\left[||V_\phi(s) - \hat{V}(s)||^2\right], \quad (14)$$

where $V_\phi(s)$ is the estimate of the value function, and $\hat{V}(s)$ is the value target computed by TD-$\lambda$ (Sutton & Barto, 2018). We follow SAPO (Xing et al., 2025) using the double-critic network and including the mean of the two value functions for computing the value target.

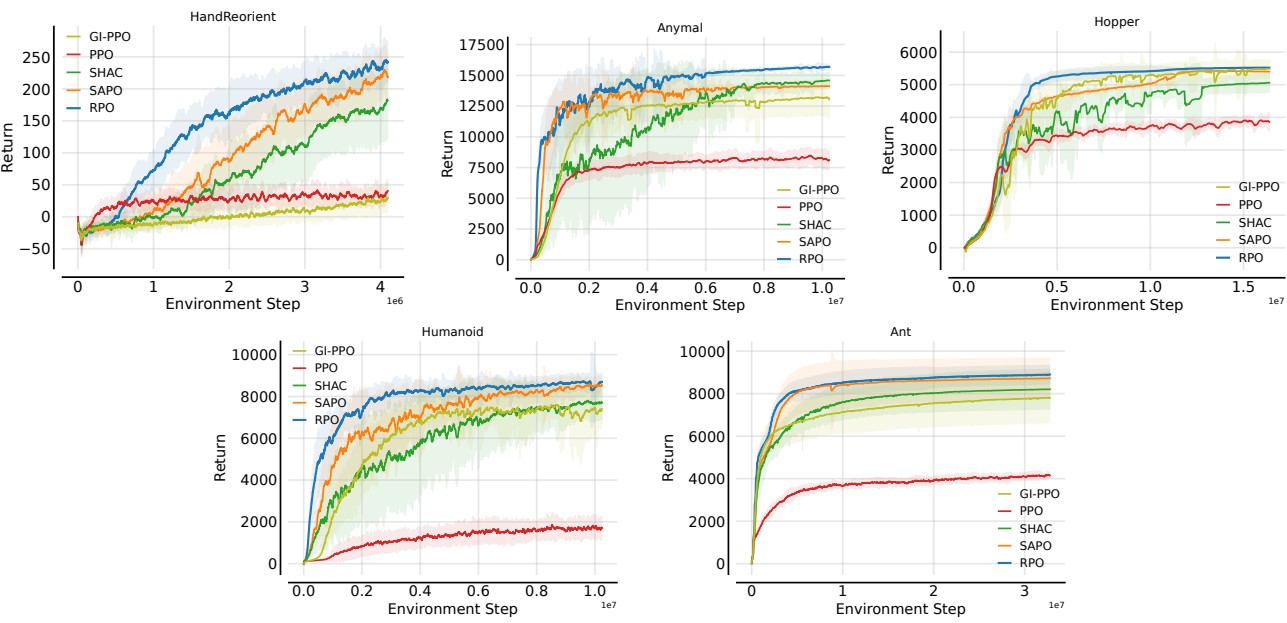

*Figure 3.* Training performance comparison of RPO, SAPO, SHAC, and PPO. Each plot shows the mean episode return over environment steps, with the shaded region representing the standard deviation. All curves are smoothed with a 100-episode moving average.

*Table 1.* Stochastic Evaluation (i.e. sampling actions from the policy distribution) for the final performance after training. Each evaluation consists of 128 episodes for each seed. Results are reported as mean ± standard deviation.

| | **Hand Reorient** | **Hopper** | **Ant** | **Humanoid** | **Anymal** |
|---|---|---|---|---|---|
| PPO | $36.83 \pm 17.50$ | $3940.60 \pm 129.72$ | $4146.00 \pm 164.14$ | $1665.66 \pm 410.93$ | $8244.86 \pm 680.87$ |
| GI-PPO | $27.19 \pm 20.58$ | $\mathbf{5512.30 \pm 285.90}$ | $7805.47 \pm 1159.23$ | $7507.85 \pm 639.52$ | $12260.46 \pm 3650.35$ |
| SHAC | $174.63 \pm 57.54$ | $5067.18 \pm 299.37$ | $8206.15 \pm 940.46$ | $7744.44 \pm 858.97$ | $14568.97 \pm 652.72$ |
| SAPO | $213.44 \pm 33.95$ | $5407.79 \pm 4.28$ | $8718.60 \pm 946.94$ | $\mathbf{8603.09 \pm 402.82}$ | $14095.90 \pm 82.02$ |
| RPO (ours) | $\mathbf{237.55 \pm 25.16}$ | $\mathbf{5525.57 \pm 3.47}$ | $\mathbf{8891.04 \pm 440.39}$ | $\mathbf{8637.78 \pm 422.17}$ | $\mathbf{15674.12 \pm 316.26}$ |

# 5. Experiments

We conduct experiments to answer the following three questions: (i) Does RPO achieve superior sample efficiency compared to previous RPG-based methods? (ii) Does RPO achieve strong performances? (iii) What are the impacts of RPO's main components on its overall performance?

## 5.1. Experimental Setup

**Environments and tasks.** We conduct experiments on a suite of five challenging continuous control tasks from two differentiable simulators, DFlex (Xu et al., 2021; Georgiev et al., 2024) and Rewarped (Xing et al., 2025). This suite is composed of four locomotion tasks and one dexterous manipulation task. The four locomotion tasks are from DFlex (Georgiev et al., 2024), where the goal is to maximize the forward velocity: (i) Hopper; (ii) Ant; (iii) Anymal; and (iv) Humanoid. The manipulation task is the Hand Reorient task from Rewarped (Xing et al., 2025), which involves an Allegro Hand learning to reorient a cube. Further details

regarding the environments are provided in Appendix A.

**Baselines.** We compare the sample efficiency and performance of RPO with SOTA RPG-based and model-free methods: (a) SAPO (Xing et al., 2025), a SOTA RPG-based method with short-horizon trajectories and entropy regularization; (b) SHAC (Xu et al., 2021), a variance reduction method for RPG, for which we use the implementation from (Xing et al., 2025) that includes several architectural changes that enhance its performance; (c) PPO (Schulman et al., 2017), a model-free policy gradient method; (d) GI-PPO (Son et al., 2023), which performs a single RPG update epoch followed by PPO-style updates for all subsequent epochs via REINFORCE. Detailed hyper-parameters and implementation specifics for all methods are provided in Appendix C.

**Metrics.** We evaluate each algorithm using 12 random seeds for each task. For main experiments, to account for simulator stochasticity, we run each seed twice (for a total of 24 runs per experiment), except in the Hopper task. Sam-

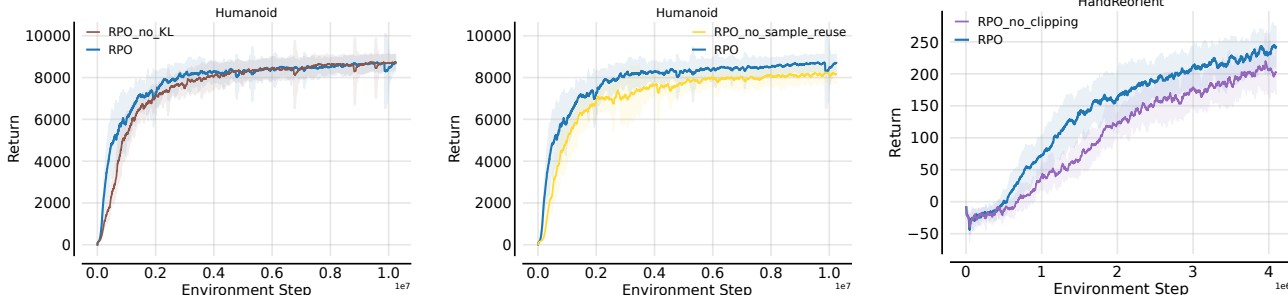

*Figure 4.* Ablation study of RPO's components. The plot shows training curves for three variants: RPO without KL regularization, RPO with only one policy update epoch (i.e, no sample reuse), and RPO without the clipping mechanism.

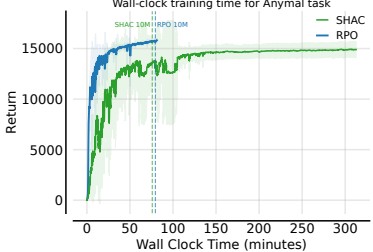

*Figure 5.* Comparison of wall-clock training time. RPO (10M environment steps) completes training in ∼ 81 minutes, compared to ∼ 313 minutes for SHAC (40M environment steps). This highlights that the slight computational overhead of sample reuse is far outweighed by the substantial boost in overall training efficiency.

ple efficiency is evaluated via training curves in Figure 3. We report the final performance over 128 episodes using both stochastic and deterministic protocols in Table 1 and Appendix B, respectively.

### 5.2. Experimental Results

**RPO achieves superior final performance.** As summarized in Table 1, RPO consistently achieves state-of-the-art (SOTA) results across all tasks. In the challenging Hand Reorient (controlling an Allegro hand for cube rotation) and ANYmal tasks (quadruped locomotion (Hutter et al., 2017)), RPO outperforms all baselines by a significant margin. Even in the Ant, Hopper, and Humanoid tasks, where baselines like SAPO and SHAC already exhibit strong performance, RPO is able to push the performance boundary further.

**RPO consistently demonstrates superior sample efficiency and stability.** RPO's efficiency stems from its ability to (i) reuse samples across multiple policy updates and (ii) stabilize RPG training via the proposed clipping mechanism and KL regularization. As shown in Figure 3, these mechanisms translate to faster learning across all tasks. For instance, in Hand Reorient task, RPO learns significantly faster than all baselines. In Hopper, RPO reaches a score of 5000 several million steps earlier than other methods. Similarly, in Ant, RPO is the fastest to reach the 8000 score.

In ANYmal , RPO surpasses the final performance of both SAPO and SHAC after only 4 million steps, and in Humanoid, it reaches a score of 8000 approximately 3 million steps faster than SAPO. Furthermore, RPO exhibits significantly higher sample efficiency compared to PPO across all environments.

**Wall-clock Time Comparison.** We investigate the computational overhead of RPO's sample reuse mechanism in terms of wall-clock time. We compare RPO (10 million environment steps) against SHAC (40 million environment steps) on the ANYmal task. Experiments were conducted on the same machine with an NVIDIA RTX 4090 GPU, with 8 seeds evaluated sequentially. As shown in Figure 5, RPO completes training in approximately 81 minutes, whereas SHAC requires roughly 313 minutes. Notably, when training for the same duration of 10 million steps, RPO takes 80 minutes compared to 76 minutes for SHAC. This comparison reveals that sample reuse introduces only a marginal computational overhead per step. Consequently, RPO's superior sample efficiency directly translates to wall-clock time efficiency, as SHAC fails to match RPO's performance even with significantly longer training.

### 5.3. Ablation Study

In our ablation study (Figure 4), we examine the contribution of each RPO component by evaluating variants without: (i) KL regularization, (ii) gradient clipping, and (iii) sample reuse. We further evaluate (iv) RPO's robustness to hyperparameter choices.

**KL regularization stabilizes policy training.** KL regularization is critical for stabilizing RPO, particularly during the early training phases. As shown in Figure 4 (a), removing KL regularization significantly slows down learning; the agent reaches a score of 8000 approximately two million steps later than the full model. Further analysis of KL regularization is provided in Appendix G.

**Clipping mechanism ensures numerical stability.** The proposed policy gradient clipping mechanism filters out

samples with large importance weight ratios. This prevents numerical instability and ensures action probabilities remain bounded. As illustrated in Figure 4 (c), removing the gradient clipping mechanism significantly degrades performance on the Hand Reorient task.

**Sample reuse improves efficiency and performance.** Sample efficiency degrades when limiting training to a single policy update epoch per iteration. Consequently, final performance is also compromised, as shown in Figure 4 (b), confirming the effectiveness of sample reuse.

**Hyperparameter sensitivity.** We test RPO's sensitivity on the ANYmal task by varying (a) the policy gradient clipping bounds $c_{low}$ and $c_{high}$, and (b) the KL regularization and entropy coefficients $\lambda_{KL}$ and $\lambda_{ent}$. RPO maintains performance comparable to the default setting under both stricter and looser clipping bounds, and across a range of KL and entropy coefficients, indicating that RPO is not brittle to precise hyperparameter choices. Detailed training curves are shown in Figure 10 of Appendix H.

## 6. Conclusion

In this work, we addressed the training instability of Reparameterization Policy Gradient by establishing a key connection between RPG and a surrogate objective. This insight provides a principled path to stable sample reuse. Based on this, we propose Reparameterization Proximal Policy Optimization, an algorithm that stabilizes policy learning by applying a tailored gradient clipping mechanism to the surrogate objective's policy gradient, further complemented by KL regularization. Our experiments on challenging locomotion and manipulation tasks confirm that RPO significantly outperforms prior methods in sample efficiency with strong performance. A promising direction for future work is investigating the sim-to-real transfer of RPO-trained policies.

## Impact Statement

This paper presents work whose goal is to advance the field of Machine Learning. There are many potential societal consequences of our work, none which we feel must be specifically highlighted here.

## Acknowledgements

This work was supported by the National Natural Science Foundation of China Grant 52494974. Hai Zhong thanks Xiaoyu Chen for helpful discussions.

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

# A. Environment and Task Details

In this section, we discuss the details of the environments and tasks used in this work. The four locomotion tasks (i.e., Anymal, Hopper, Ant, and Humanoid) are from the DFlex simulator (Xu et al., 2021; Georgiev et al., 2024). Specifically, we use the versions from AHAC's official implementation (`https://github.com/imgeorgiev/DiffRL`). All locomotion tasks aim to learn a policy that maximizes the agent's forward velocity. The Hand Reorient task is from the official implementation of Rewarped (`https://github.com/rewarped/rewarped`), version 1.3.0.

### A.1. Ant

Ant ($S \in \mathbb{R}^{37}, A \in \mathbb{R}^8$) is a four-legged robot. The reward function is defined as (Georgiev et al., 2024):

$$v_x + R_{height} + 0.1R_{angle} + R_{heading} - 0.01\|a\|^2,$$

where $v_x$ is the forward velocity, and the other components are: $R_{height}$, which encourages the robot to stand up; $R_{angle}$, which rewards an upward-pointing normal vector; $R_{heading}$, which promotes forward movement; and a penalty on the action norm, $-0.01\|a\|^2$, to encourage energy-efficient policies.

### A.2. Anymal

Anymal ($S \in \mathbb{R}^{49}, A \in \mathbb{R}^{12}$) is a real quadrupedal robot (Hutter et al., 2017). The reward function is defined as (Georgiev et al., 2024):

$$v_x + R_{height} + 0.1R_{angle} + R_{heading} - 0.01\|a\|^2.$$

### A.3. Hopper

Hopper ($S \in \mathbb{R}^{11}, A \in \mathbb{R}^3$) is a three-jointed planar robot. The reward function is defined as (Georgiev et al., 2024):

$$v_x + R_{height} + R_{angle} - 0.1\|a\|^2.$$

### A.4. Humanoid

Humanoid ($S \in \mathbb{R}^{76}, A \in \mathbb{R}^{21}$) is a high-dimensional bipedal robot. The reward function is defined as (Georgiev et al., 2024):

$$v_x + R_{height} + 0.1R_{angle} + R_{heading} - 0.02\|a\|^2.$$

### A.5. Hand Reorient

This task involves an Allegro Hand ($S \in \mathbb{R}^{72}, A \in \mathbb{R}^{16}$) learning to reorient a cube to a target pose. This task was adapted for Rewarped (Xing et al., 2025) from Isaac Gym (Makoviychuk et al., 2021). The detailed reward function can be found in the original Isaac Gym paper (Makoviychuk et al., 2021).

# B. Deterministic Evaluation Results

In this section, we provide the deterministic evaluation results for the five tasks used in the paper. For stochastic evaluations, we sample actions from the policy distribution. Results are shown in Table 6. As shown in Table 6, RPO achieves best results across tasks.

# C. Hyperparameters and Implementation Details

### C.1. Hyperparameters and Architectures

We detail the hyperparameters and architectures used for all algorithms in Table 2. For our GI-PPO baseline, we followed the official implementation (`https://github.com/SonSang/gippo`) but made several improvements. These changes include using a double critic architecture, switching from the Adam to the AdamW optimizer, aligning the network size with our method, and changing the activation function from ELU to SiLU. Since GI-PPO is sensitive to its hyperparameters, we performed an extensive search within our computational budget. The resulting hyperparameters

*Table 2.* Common hyperparameters for all algorithms.

|  | *shared* | PPO | SHAC | SAPO | RPO | GI-PPO |
|---|---|---|---|---|---|---|
| Horizon $h$ | 32 | | | | | |
| Epochs for critics $L$ | | 5 | 16 | 16 | 16 | 16 |
| Epochs for actors $M$ | | 5 | 1 | 1 | 5 | 6 |
| Discount $\gamma$ | 0.99 | | | | | |
| TD/GAE $\lambda$ | 0.95 | | | | | |
| Actor MLP | $(400, 200, 100)$ | shared actor-critic MLP | | | | |
| Critic MLP | $(400, 200, 100)$ | shared actor-critic MLP | | | | |
| Actor $\eta$ | | $5e-4$ | $2e-3$ | $2e-3$ | $5e-4$ | $5e-4$ |
| Critic $\eta$ | $5e-4$ | | | | | |
| Entropy $\eta$ | - | | | $5e-3$ | | |
| $\eta$ schedule | - | KL(0.008) | linear | linear | exponential | N.A. |
| Optim type | AdamW | | | | | |
| Optim $(\beta_1, \beta_2)$ | $(0.7, 0.95)$ | $(0.9, 0.999)$ | | | | |
| Grad clip | 0.5 | | | | | 1.0 |
| Norm type | LayerNorm | | | | | |
| Activation type | SiLU | | | | | |
| Num critics $C$ | - | | 2 | 2 | 2 | 2 |
| Target entropy $\bar{\mathcal{H}}$ | - | | | $-\dim(\mathcal{A})/2$ | $-\dim(\mathcal{A})/2$ | |
| Init temperature | - | | | 1.0 (0.005 for Hand Reorient) | | |

*Table 3.* The number of parallel environments used for each environment. These values are kept the same as in the official implementations: we follow the AHAC repository (https://github.com/imgeorgiev/DiffRL) for the DFlex tasks and the Rewarped repository (https://github.com/rewarped/rewarped) for the Hand Reorient task.

|  | Hopper | Ant | Humanoid | Anymal | Hand Reorient |
|---|---|---|---|---|---|
| Num Envs | 1024 | 128 | 64 | 128 | 64 |

*Table 4.* RPO's unique hyperparameters.

|  | Hopper | Ant | Humanoid | Anymal | Hand Reorient |
|---|---|---|---|---|---|
| Entropy coefficient | 0.25 | 0.2 | 0.5 | 0.25 | 0.001 |
| KL coefficient | 0.2 | 0.25 | 0.5 | 0.2 | 0.003 |
| $c_{low}$ | 0.8 | 0.8 | 0.8 | 0.8 | 0.8 |
| $c_{high}$ | 1.0 | 1.0 | 1.0 | 1.0 | 1.0 |

*Table 5.* GI-PPO's unique hyperparameters.

|  | Hopper | Ant | Humanoid | Anymal | Hand Reorient |
|---|---|---|---|---|---|
| alpha | $5e-1$ | $5e-1$ | $5e-3$ | $1e-3$ | $5e-4$ |
| max oorr | 0.7 | 0.8 | 0.1 | 0.5 | 0.1 |
| e clip | 0.2 | 0.2 | 0.2 | 0.2 | 0.05 |
| alpha interval | 0.4 | 0.4 | 0.4 | 0.4 | 0.4 |
| alpha update factor | 1.02 | 1.02 | 1.02 | 1.02 | 1.02 |

*Table 6.* Deterministic Evaluation for the final performance after training. Each evaluation consists of 128 episodes. Mean and standard deviation.

|  | **Hand Reorient** | **Hopper** | **Ant** | **Humanoid** | **Anymal** |
|---|---|---|---|---|---|
| PPO | $37.18 \pm 12.76$ | $3977.85 \pm 159.95$ | $4339.51 \pm 745.48$ | $2140.61 \pm 529.07$ | $10257.12 \pm 2247.05$ |
| GI-PPO | $36.78 \pm 20.90$ | $\mathbf{5514.00 \pm 285.64}$ | $7812.05 \pm 1165.10$ | $7576.97 \pm 621.58$ | $12247.36 \pm 3552.17$ |
| SHAC | $175.13 \pm 55.10$ | $5068.42 \pm 299.73$ | $8205.95 \pm 940.18$ | $7722.20 \pm 742.96$ | $14560.59 \pm 655.30$ |
| SAPO | $225.17 \pm 27.66$ | $5478.12 \pm 4.45$ | $\mathbf{9101.36 \pm 996.14}$ | $8676.25 \pm 420.62$ | $14783.27 \pm 53.01$ |
| **RPO (ours)** | $\mathbf{239.70 \pm 19.77}$ | $\mathbf{5584.86 \pm 6.41}$ | $9072.29 \pm 448.57$ | $\mathbf{8805.61 \pm 361.59}$ | $\mathbf{15872.61 \pm 461.15}$ |

used are listed in Table 5. Our implementations of SAPO, PPO, and SHAC are based on the official SAPO repository (`https://github.com/etaoxing/mineral`). Most hyperparameters are kept consistent with that repository, with a few key exceptions for fair comparison: the number of parallel environments and the MLP size are aligned with the official AHAC repository (Georgiev et al., 2024). To ensure a fair comparison, most hyperparameters and the core architecture are shared across all tested algorithms. We tuned the initial temperature for SAPO in the Hand Reorient task, as the default setting (i.e, in SAPO's paper for this task) of 1.0 from the SAPO paper was found to be too high. Specifically for SHAC, we use the improved version from the SAPO repository, which aligns its architecture with that of RPO and SAPO. Our RPO implementation is also built upon the SAPO repository.

As stated in the main text, our main experiments use 12 random seeds per algorithm with two simulator instantiations per seed (24 runs per algorithm per task). Due to our computational budget, for some ablations whose performance gaps are sufficiently large to be conclusive, we use 12 runs (one simulator instantiation per seed) instead of 24; the remaining ablations follow the same 24-run protocol as the main experiments.

### C.2. Implementation Details for RPO

We use the squashed normal policy class for all RPG methods. Though the derivations of RPG are for Gaussian policies, it also works equally well for squashed normal policies.

For the critic, we use double critic and mean average as target for TD training, following (Xing et al., 2025). For the entropy regularization, we follow SAPO to add an entropy bonus to the reward, which is scaled by a target entropy (Xing et al., 2025). But note that the gradients of entropy are not backpropagated to actions at the same time step (they backpropagate to the policy parameters directly, while the entropy gradients could flow through states to previous actions and we do not explicitly filter this gradient path out), so we calculate the gradients of entropy explicitly with respect to the policy parameter during policy updates. We do not discount the explicitly calculated entropy gradients, and this works empirically well. We retain the entropy reward, since it smooths critic training.

## D. Details for Connecting RPG and Surrogate Objective

In this section, we give details to explain the connection between RPG and the surrogate objective. First, we rewrite (8) by expanding $d^{\pi_{\theta_{\text{old}}}}(s)$ and interchanging the order of integration and summation:

$$
\begin{aligned}
\nabla_\theta L_{\pi_{\theta_{\text{old}}}}(\theta) &= \int_s d^{\pi_{\theta_{\text{old}}}}(s) \int_\epsilon \left[ \nabla_\theta a \nabla_a Q^{\pi_{\theta_{\text{old}}}}(s,a)|_{a=f_\theta(\epsilon;s)} p_{\text{std}}(\epsilon) \mathrm{d}\epsilon \right] \mathrm{d}s, \\
&= \int_s \sum_{t=0}^\infty \gamma^t p(s_t = s | \pi_{\theta_{\text{old}}}) \int_\epsilon \left[ \nabla_\theta a \nabla_a Q^{\pi_{\theta_{\text{old}}}}(s,a)|_{a=f_\theta(\epsilon;s)} p_{\text{std}}(\epsilon) \mathrm{d}\epsilon \right] \mathrm{d}s, \\
&= \sum_{t=0}^\infty \int_s \gamma^t p(s_t = s | \pi_{\theta_{\text{old}}}) \int_\epsilon \left[ \nabla_\theta a \nabla_a Q^{\pi_{\theta_{\text{old}}}}(s,a)|_{a=f_\theta(\epsilon;s)} p_{\text{std}}(\epsilon) \mathrm{d}\epsilon \right] \mathrm{d}s,
\end{aligned}
\tag{15}
$$

which is the policy gradient to be estimated and $p_{\text{std}}(\epsilon)$ is the probability density function for standard Gaussian distribution.

The derivation above involves interchanging summation over time steps and integration. Under regularity conditions, this interchange is valid. When these conditions are not satisfied, we can instead consider a truncated-horizon variant: as long as the per-timestep gradient contribution is well-defined (which is a minimum requirement for the reparameterization gradient to be meaningful at all—if even a single time step's contribution diverges, the method itself becomes ill-posed), the truncated estimator with horizon $T$ is an unbiased estimate of the truncated surrogate gradient via finite-sum linearity for finite $T$, and serves as a good practical approximation for policy optimization when $T$ is sufficiently large. Empirically, irrespective of whether these regularity conditions are precisely satisfied, the resulting algorithm performs robustly across a range of differentiable simulation tasks, suggesting that the proposed methods are effective in practical settings.

### D.1. Collect Rollouts and Compute Action-gradients

First, as shown in Section 4.1.1, we collect a batch of rollouts and compute the gradients of discounted cumulative return with respect to the action at each time step with BPTT. From here, we clearly see that the action-gradient for time step $k$, $\gamma^k \nabla_{a_k} \sum_{t=k}^\infty \gamma^{(t-k)} r(s_t, a_t)$, is exactly an unbiased Monte Carlo estimate of $\gamma^k \nabla_a Q^{\pi_{\theta_{\text{old}}}}(s_k, a_k)$, where $s_k$ and $a_k$ are sampled according to $p(s_k = s | \pi_{\theta_{\text{old}}})$ and $\pi_{\theta_{\text{old}}}(a|s)$. We cache these action-gradients for further calculations.

## D.2. On-Policy Gradient

For the first policy update (on-policy update), the behavior policy $\pi_{\theta_{old}}$ is identical to the policy $\pi_\theta$ being updated. The on-policy update is a special case of the off-policy update, in which the importance weight ratio collapses to 1. We describe the more general off-policy procedure in detail in the following subsection.

## D.3. Off-Policy Gradient

After the first policy update, $\pi_\theta$ is updated to $\pi_{\theta'}$, and we need to compute off-policy gradients. Throughout this subsection, $\epsilon_{old}$ denotes the source noise of the behavior policy $\pi_{\theta_{old}}$, sampled as $\epsilon_{old} \sim p_{std}$, while $\epsilon$ denotes the source noise of the updated policy $\pi_{\theta'}$. To do so, we need to account for the fact that to regenerate the same action collected in the rollout, a specific value of $\epsilon$ is required for $\theta'$, generally different from $\epsilon_{old}$.

To generate the sampled action $a_k$, $\epsilon$ and $\epsilon_{old}$ are linked by the following relation:

$$a_k = f_{\theta_{old}}(\epsilon_{old}; s_k) = f_{\theta'}(\epsilon; s_k). \tag{16}$$

Since we consider reparameterization Gaussian transformations in this paper, $f_{\theta_{old}}$ and $f_{\theta'}$ are invertible. We can therefore compute $\epsilon = f_{\theta'}^{-1}(a_k; s_k)$ and then regenerate $a_k$ with the updated policy $\pi_{\theta'}$. Now, we backpropagate cached action-gradients to policy network parameters $\theta'$ and obtain an estimate of $\gamma^k \nabla_{\theta'} a_k \nabla_{a_k} Q^{\pi_{\theta_{old}}}(s_k, a_k)|_{a_k=f_{\theta'}(\epsilon;s_k)}$.

However, instead of sampling $\epsilon$ directly from $\mathcal{N}(0, \mathcal{I})$, we sample it as $\epsilon = f_{\theta'}^{-1}(f_{\theta_{old}}(\epsilon_{old}; s_k); s_k)$, where $\epsilon_{old}$ is sampled from $\mathcal{N}(0, \mathcal{I})$. Hence, the probability density $p_{reg}(\epsilon|s_k)$ is different from standard Gaussian. We therefore multiply the computed gradient term $\gamma^k \nabla_{\theta'} a_k \nabla_{a_k} Q^{\pi_{\theta_{old}}}(s_k, a_k)|_{a_k=f_{\theta'}(\epsilon;s_k)}$ by the importance ratio $\rho(\theta') = \frac{\pi_{\theta'}(a_k|s_k)}{\pi_{\theta_{old}}(a_k|s_k)}$ as an off-policy correction. We now give a formal proof that this importance ratio is the correct correction: specifically, the IS-weighted expectation matches the target time-step contribution $\int_s \gamma^k p(s_k = s|\pi_{\theta_{old}}) \int_\epsilon \left[ \nabla_{\theta'} a \nabla_a Q^{\pi_{\theta_{old}}}(s, a)|_{a=f_{\theta'}(\epsilon;s)} p_{std}(\epsilon) \right] d\epsilon ds$.

**Proposition D.1.** *Let the state $s_k$ be sampled from the state distribution of the behavior policy, $s_k \sim p(s_k = s|\pi_{\theta_{old}})$. Let the reparameterization noise $\epsilon_{old}$ be sampled from the standard Normal distribution, $\epsilon_{old} \sim p_{std}(\epsilon_{old})$, and define the regenerated noise as $\epsilon = f_{\theta'}^{-1}(f_{\theta_{old}}(\epsilon_{old}; s_k); s_k)$, whose induced distribution is denoted $p_{reg}(\epsilon|s_k)$, where $f_{\theta'}$ and $f_{\theta_{old}}$ are reparameterization Gaussian transformations as shown in this paper.*

*Define the off-policy gradient estimator $G(\theta')$ as the random variable:*

$$G(\theta') = \gamma^k \frac{\pi_{\theta'}(a_k|s_k)}{\pi_{\theta_{old}}(a_k|s_k)} \nabla_{\theta'} a_k \nabla_{a_k} Q^{\pi_{\theta_{old}}}(s_k, a_k)|_{a_k=f_{\theta'}(\epsilon;s_k)}$$

*where the action $a_k = f_{\theta_{old}}(\epsilon_{old}; s_k) = f_{\theta'}(\epsilon; s_k)$.*

*Then, its expectation is given by:*

$$\mathbb{E}_{\substack{s_k \sim p(\cdot|\pi_{\theta_{old}}) \\ \epsilon \sim p_{reg}(\cdot|s_k)}} [G(\theta')] = \int_s \gamma^k p(s_k = s|\pi_{\theta_{old}}) \int_\epsilon \left[ \nabla_{\theta'} a \nabla_a Q^{\pi_{\theta_{old}}}(s_k, a)|_{a=f_{\theta'}(\epsilon;s_k)} p_{std}(\epsilon) \right] d\epsilon ds.$$

*Proof:* Consider the Gaussian reparameterization $f_\theta(\epsilon; s_k) = \mu_\theta(s_k) + \sigma_\theta(s_k) \epsilon$, where $\sigma_\theta(s_k)$ is a diagonal matrix with strictly positive diagonal entries and is therefore invertible. Throughout this proof, $Df(x; s)$ denotes the Jacobian matrix of $f$ with respect to the first argument $x$, evaluated at the given point.

**Full support of $\epsilon$.** By definition, $a_k = f_{\theta_{old}}(\epsilon_{old}; s_k) = f_{\theta'}(\epsilon; s_k)$, i.e., $\mu_{\theta_{old}}(s_k) + \sigma_{\theta_{old}}(s_k) \epsilon_{old} = \mu_{\theta'}(s_k) + \sigma_{\theta'}(s_k) \epsilon$. Solving for $\epsilon$ directly yields the explicit affine relation

$$\epsilon = \sigma_{\theta'}^{-1}(s_k) \sigma_{\theta_{old}}(s_k) \epsilon_{old} + \sigma_{\theta'}^{-1}(s_k) \left( \mu_{\theta_{old}}(s_k) - \mu_{\theta'}(s_k) \right).$$

Since $\sigma_{\theta'}^{-1}(s_k)\sigma_{\theta_{old}}(s_k)$ is a positive diagonal matrix and thus invertible, the mapping from $\epsilon_{old}$ to $\epsilon$ is an affine bijection on $\mathbb{R}^d$. Because $\epsilon_{old}$ has full support on $\mathbb{R}^d$, so does $\epsilon$, matching the support of $p_{std}$.

**Density of $\epsilon$.** By the multivariate change-of-variable formula (Grimmett & Stirzaker, 2001) applied to $\epsilon_{old} = f_{\theta_{old}}^{-1}(f_{\theta'}(\epsilon; s_k); s_k)$ together with the chain rule,

$$p_{reg}(\epsilon|s_k) = p_{std}(\epsilon_{old}) \left| \det Df_{\theta_{old}}^{-1}(a_k; s_k) \right| \left| \det Df_{\theta'}(\epsilon; s_k) \right|.$$

**Importance weight ratio.** The Gaussian policy densities admit the change-of-variable form

$$\pi_{\theta_{\text{old}}}(a_k|s_k) = p_{\text{std}}(\epsilon_{\text{old}}) \left| \det Df_{\theta_{\text{old}}}^{-1}(a_k; s_k) \right|, \qquad \pi_{\theta'}(a_k|s_k) = p_{\text{std}}(\epsilon) \left| \det Df_{\theta'}^{-1}(a_k; s_k) \right|,$$

so

$$\frac{\pi_{\theta'}(a_k|s_k)}{\pi_{\theta_{\text{old}}}(a_k|s_k)} = \frac{p_{\text{std}}(\epsilon) \left| \det Df_{\theta'}^{-1}(a_k; s_k) \right|}{p_{\text{std}}(\epsilon_{\text{old}}) \left| \det Df_{\theta_{\text{old}}}^{-1}(a_k; s_k) \right|}.$$

**Cancellation.** Multiplying the density by the importance ratio,

$$\frac{\pi_{\theta'}(a_k|s_k)}{\pi_{\theta_{\text{old}}}(a_k|s_k)} \, p_{\text{reg}}(\epsilon|s_k) = p_{\text{std}}(\epsilon) \underbrace{\left| \det Df_{\theta'}(\epsilon; s_k) \right| \left| \det Df_{\theta'}^{-1}(a_k; s_k) \right|}_{= 1} = p_{\text{std}}(\epsilon),$$

where the underbraced product equals one by the inverse function theorem, as $f_{\theta'}(\cdot; s_k)$ and $f_{\theta'}^{-1}(\cdot; s_k)$ are mutual inverses at $\epsilon$ and $a_k$.

**Conclusion.** Expanding the proposition's left-hand side using the joint density $p(s_k, \epsilon) = p(s_k = s|\pi_{\theta_{\text{old}}}) \cdot p_{\text{reg}}(\epsilon|s_k)$:

$$\mathbb{E}_{\substack{s_k \sim p(\cdot|\pi_{\theta_{\text{old}}}) \\ \epsilon \sim p_{\text{reg}}(\cdot|s_k)}} [G(\theta')]$$

$$= \int_s p(s_k = s|\pi_{\theta_{\text{old}}}) \int_\epsilon \gamma^k \frac{\pi_{\theta'}(a_k|s_k)}{\pi_{\theta_{\text{old}}}(a_k|s_k)} \nabla_{\theta'} a \, \nabla_a Q^{\pi_{\theta_{\text{old}}}}(s_k, a) \Big|_{a = f_{\theta'}(\epsilon; s_k)} p_{\text{reg}}(\epsilon|s_k) \, d\epsilon \, ds$$

$$= \int_s \gamma^k p(s_k = s|\pi_{\theta_{\text{old}}}) \int_\epsilon \nabla_{\theta'} a \, \nabla_a Q^{\pi_{\theta_{\text{old}}}}(s_k, a) \Big|_{a = f_{\theta'}(\epsilon; s_k)} p_{\text{std}}(\epsilon) \, d\epsilon \, ds,$$

where the second equality substitutes the cancellation identity $\rho(\theta') \cdot p_{\text{reg}}(\epsilon|s_k) = p_{\text{std}}(\epsilon)$ into the inner integral. This is the proposition's right-hand side, completing the proof. $\qquad\square$

The estimator $G(\theta')$ in Proposition D.1 contains the term $\gamma^k \nabla_a Q^{\pi_{\theta_{\text{old}}}}(s_k, a_k)$, a theoretical quantity not directly accessible from a single rollout. In practice, we replace it with the unbiased pathwise estimator $\nabla_{a_k} R(\tau)$ (see Section 4.1.1). Conditional on $(s_k, \epsilon)$ (and therefore $a_k = f_{\theta'}(\epsilon; s_k)$), the future of $\tau$ beyond time step $k$ is determined by the post-$k$ behavior noises $\epsilon_{\text{old}, >k} \sim p_{\text{std}}$, independent of $(s_k, \epsilon)$ and the history. Define the trajectory-level estimator

$$G_{\text{traj}}(\theta'; s_k, \epsilon, \epsilon_{\text{old}, >k}) := \frac{\pi_{\theta'}(a_k|s_k)}{\pi_{\theta_{\text{old}}}(a_k|s_k)} \nabla_{\theta'} a_k \, \nabla_{a_k} R(\tau) \Big|_{a_k = f_{\theta'}(\epsilon; s_k)}.$$

By the tower property,

$$\mathbb{E}_{s_k, \epsilon, \epsilon_{\text{old}, >k}} \big[ G_{\text{traj}} \big] = \mathbb{E}_{s_k, \epsilon} \Big[ \mathbb{E}_{\epsilon_{\text{old}, >k}} \big[ G_{\text{traj}} \, \big| \, s_k, \epsilon \big] \Big] = \mathbb{E}_{s_k, \epsilon} \big[ G(\theta') \big],$$

where the inner expectation collapses to $G(\theta')$ because $\mathbb{E}_{\epsilon_{\text{old}, >k}} [\nabla_{a_k} R(\tau) \, | \, s_k, \epsilon] = \gamma^k \nabla_a Q^{\pi_{\theta_{\text{old}}}}(s_k, a_k)$ (Section 4.1.1).

Based on this result, by summing over time steps along each trajectory and averaging across trajectories, we obtain a Monte Carlo estimate of the off-policy reparameterization gradient for the surrogate objective. Rigorously, we cannot sum over time steps to infinity in practice; taking the horizon $T$ to be sufficiently large (or, as in our implementation, using short-horizon rollouts with value-function bootstrapping) yields an accurate estimate.

**Summary of practical approximations.** In theory, the infinite-horizon series form of the surrogate policy gradient is

$$\nabla_\theta L_{\pi_{\theta_{\text{old}}}}(\theta) = \sum_{t=0}^\infty \int_s \gamma^t p(s_t = s|\pi_{\theta_{\text{old}}}) \int_\epsilon \Big[ \nabla_\theta a \nabla_a Q^{\pi_{\theta_{\text{old}}}}(s, a) \big|_{a = f_\theta(\epsilon; s)} p_{\text{std}}(\epsilon) \Big] d\epsilon \, ds,$$

which involves both an outer summation $\sum_{t=0}^\infty$ over time steps and a per-step infinite-horizon action-gradient $\nabla_{a_k} R(\tau)$. In practice, neither can be computed exactly. We use a unified mechanism to handle both: finite-horizon rollouts with value-function bootstrapping. Specifically, we approximate the infinite-horizon series above by truncating the outer summation to a finite horizon $T$ and replacing $\nabla_{a_k} R(\tau)$ by its finite-horizon counterpart that uses the value function $V$ to estimate the truncated tail return. With sufficiently large $T$ and an accurate value function, the resulting estimator is a practical and valid approximation of the infinite-horizon target.

# E. Ablation Experiments on Learning Rate

RPO utilizes an exponential learning rate decay schedule. This strategy leverages RPO's high sample efficiency for rapid initial learning, while the decaying rate accelerates final convergence. We investigate the effect of applying this exact same exponential schedule to SAPO. We evaluate two variants of SAPO with initial actor learning rates of $5 \times 10^{-4}$ and $2 \times 10^{-3}$ (SAPO's default). As shown in Figure 6, RPO consistently outperforms SAPO under both settings.

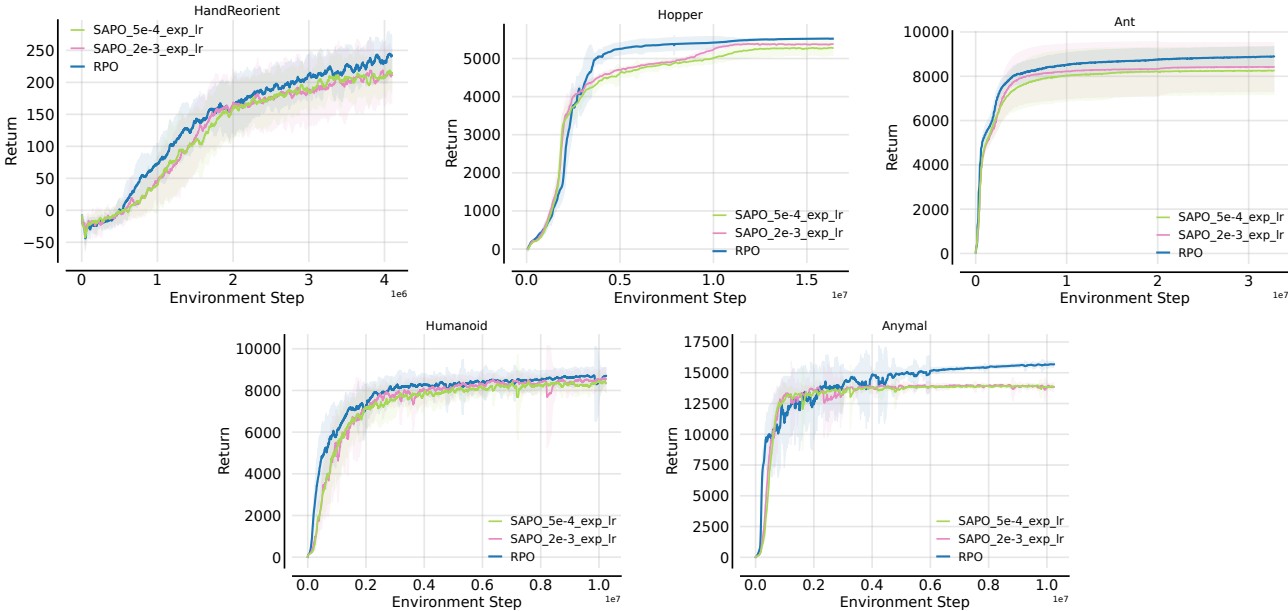

*Figure 6.* Ablation study comparing RPO and SAPO with an exponential learning rate schedule. We evaluate SAPO with initial actor learning rates of $5\mathrm{e}{-4}$ and $2\mathrm{e}{-3}$ (default setting).

# F. More Comparisons with Baselines

## F.1. Comparison with Soft Actor-Critic

We compare RPO with Soft Actor-Critic (SAC) (Haarnoja et al., 2018), incorporating the n-step return mechanism for critic training, on two tasks: Anymal and Humanoid. We utilize the implementation from the Mineral repository (https://github.com/etaoxing/mineral) and align the actor and critic MLP hidden layers to $[400, 200, 100]$. We tuned SAC's hyperparameters, including the n-step horizon, target critic smoothing coefficient, and actor learning rate, as detailed in Table 7. The training curves are presented in Figure 7. The results demonstrate that RPO consistently outperforms SAC.

*Table 7.* Hyperparameters for SAC.

|  | Humanoid | Anymal |
| --- | --- | --- |
| n-step | 3 | 10 |
| actor learning rate | $5e-4$ | $2e-3$ |
| target critic smoothing coefficient | 0.2 | 0.6 |

## F.2. Comparison with PPO Trained on More Samples

It is well known that PPO, due to its reliance on REINFORCE-type policy gradients, is much less sample-efficient than RPG-based approaches (Mohamed et al., 2020; Xu et al., 2021). We trained PPO on the Hopper task with 200 million environment steps and on the Anymal task with 100 million environment steps. The results show that even with 10 times more interactions with the environment, RPO still achieves a higher reward than PPO. The training curves are shown in Figure 8.

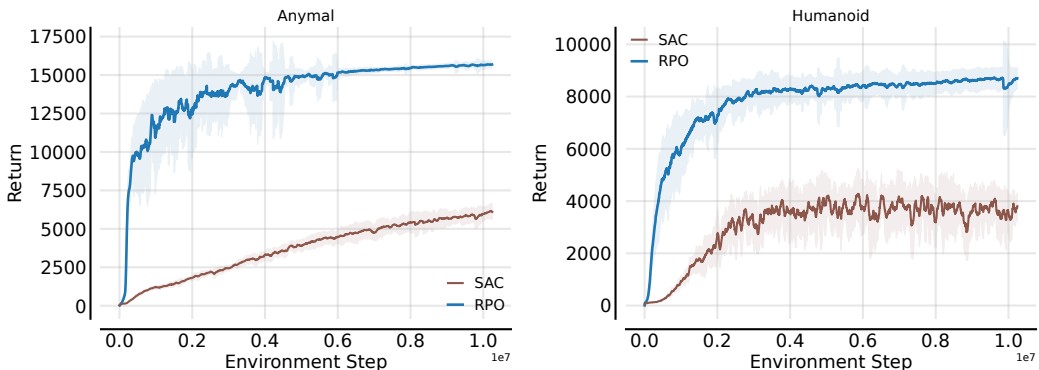

*Figure 7.* Ablation study for comparison of RPO and SAC.

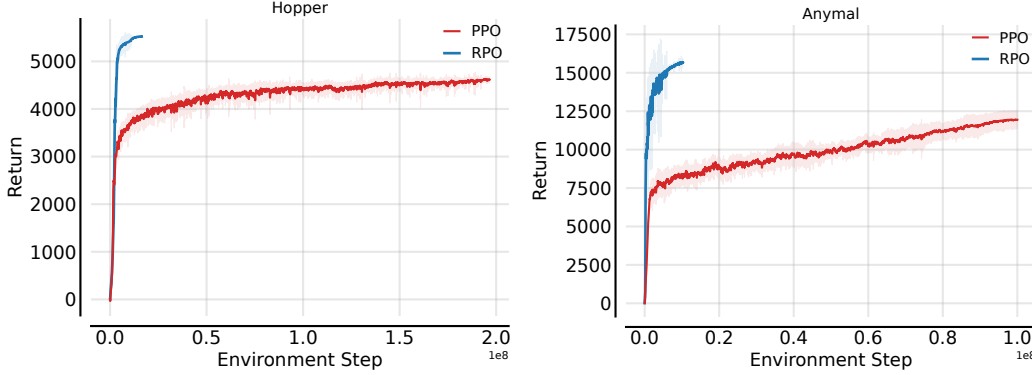

*Figure 8.* Comparison between RPO and PPO trained on significantly more samples.

# G. More Ablation on KL Divergence Regularization

We conducted further ablation studies on RPO without KL divergence regularization on the Hopper task to isolate its impact. The training curves are presented in Figure 9. As discussed in the main text, RPO without KL regularization learns noticeably more slowly than standard RPO. This retardation is primarily due to unstable policy updates.

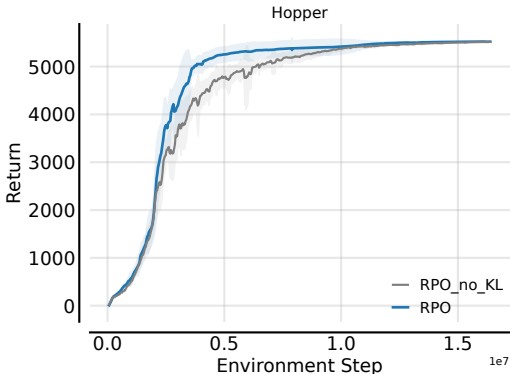

Figure 9. Ablation study for RPO without KL regularization on the Hopper task.

# H. Hyperparameter Sensitivity Analysis

This appendix complements the hyperparameter robustness analysis in Section 5.3 by providing the detailed training curves. Figure 10 reports the curves on the ANYmal task for variations of (i) the policy gradient clipping bounds $c_{low}$ and $c_{high}$, and (ii) different combinations of KL regularization ($\lambda_{KL}$) and entropy ($\lambda_{ent}$) coefficients.

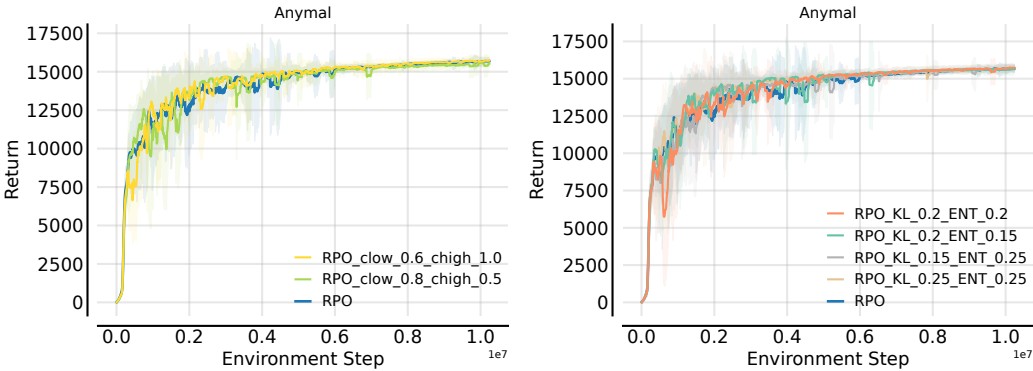

Figure 10. Ablation study on clipping values and KL/entropy coefficients.

# I. Discussion on the Design Choices of KL Regularization and Policy Gradient Clipping

In this section, we discuss the rationale behind using both KL regularization and the policy gradient clipping mechanism to stabilize policy training.

**The clipping mechanism alone is insufficient to stabilize training, whereas KL regularization stabilizes training reliably.** To demonstrate this, we conducted experiments on the Humanoid task for RPO without the KL loss, using stricter clipping settings of $c_{low} = 0.1, c_{high} = 0.1$ and $c_{low} = 0.2, c_{high} = 0.2$, respectively. As shown in Figure 11, even with small clipping ranges, policy updates remain unstable. On the other hand, KL regularization allows us to explicitly regularize policy updates, leading to stable updates while enabling maximal and stable sample reuse.

**The clipping mechanism is still necessary alongside KL regularization to filter out large importance weight ratios.** Importance weight ratios can be large in practice. Hence, it is natural to incorporate a gradient clipping mechanism to prevent policy updates driven by extreme importance weight ratios and to prevent the probability ratio of certain actions from becoming too low. Additionally, since we must calculate the importance weight ratio for unbiased policy gradient estimation regardless, there is minimal computational overhead in incorporating the policy gradient clipping mechanism.

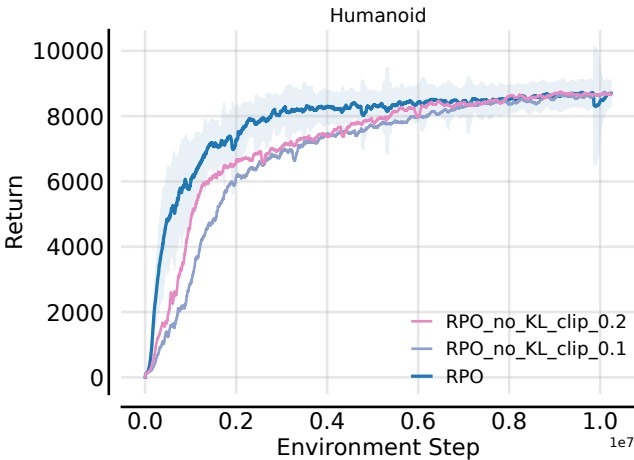

*Figure 11.* Comparison of return between RPO and RPO without KL regularization (using stricter clipping values).

## J. More Examples for Instability of RPG-based Methods

In this section, we show examples of unstable seeds for SAPO and SHAC, which are summarized in Figure 12 and Figure 13.

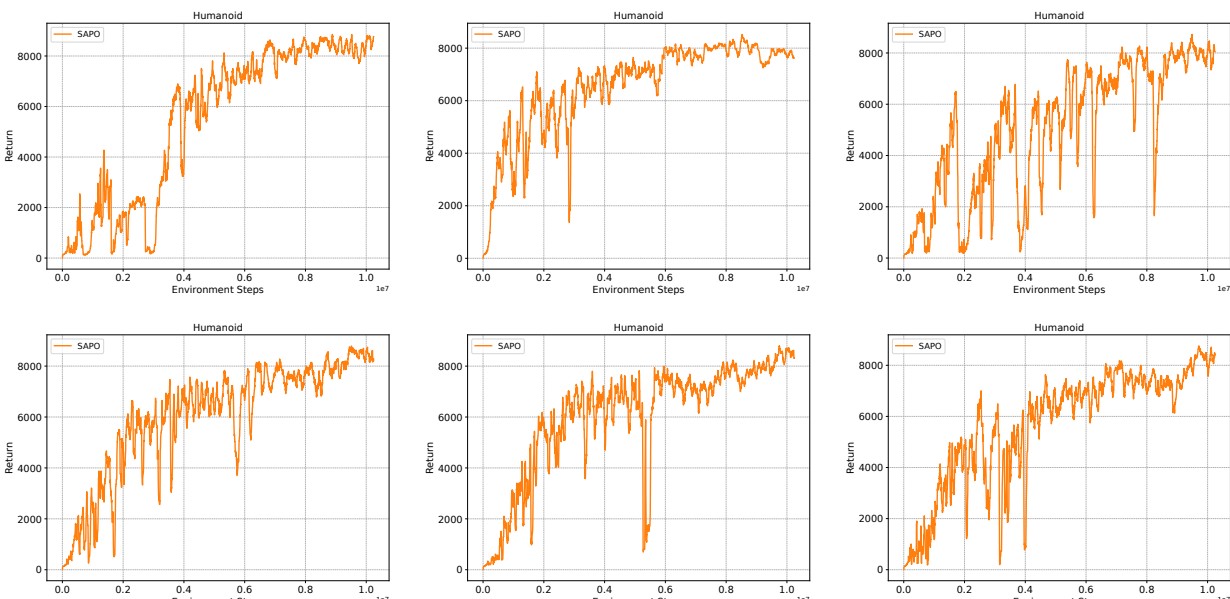

*Figure 12.* Unstable Seeds for SAPO on the Humanoid tasks.

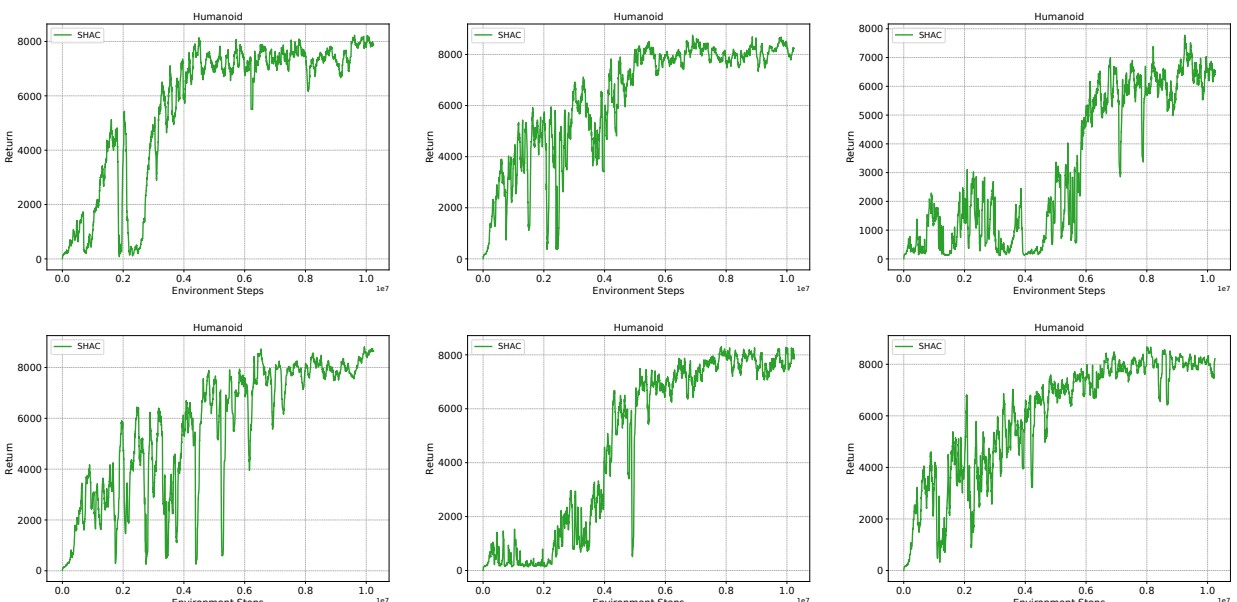

*Figure 13.* Unstable Seeds for SHAC on the Humanoid tasks.

