# OpenReview forum: "Reparameterization Proximal Policy Optimization"
_ICML.cc/2026/Conference — ICML 2026 regular_

### Official Review · Reviewer_9iea · 2026-03-11

**Soundness:** 3
**Presentation:** 2
**Significance:** 2
**Originality:** 2
**Overall Recommendation:** 4
**Confidence:** 4

**Summary:**

This paper proposes RPO, a method that enables sample reuse in Reparameterization Policy Gradient by establishing a connection between RPG and a PPO-style surrogate objective. The main technical insight is that cached action-gradients from BPTT can be reused across multiple policy updates via importance weighting. The paper adds a custom clipping mechanism and KL regularization to stabilize training.

**Compliance With Llm Reviewing Policy:**

Affirmed.

**Final Justification:**

The rebuttal has addressed my main concerns, and I increased my score.

**Key Questions For Authors:**

Can you provide experiments on tasks with discontinuous/stiff dynamics where RPG typically struggles?

What are the specific values for the clipping bounds Clow and Chigh, and how sensitive are results to these choices?

Could you include key hyperparameter sensitivity results (from Appendix I) in the main paper for completeness?

**Limitations:**

Theoretical contribution is shallow. Clipping and KL mechanisms are heuristic. Only 5 tasks from 2 simulators - no "stiff" contact dynamics where RPG typically fails.

**Strengths And Weaknesses:**

The core insight connecting RPG to the surrogate objective is genuinely interesting. The observation that action-gradients from BPTT can be viewed as Monte Carlo estimates of Q-function gradients (and thus reused with importance weighting) is a nice unifying perspective.

The paper addresses a real practical limitation - the waste of expensive dynamics Jacobians in current on-policy RPG methods. For sim-to-real applications where data is precious, this matters.

My concerns:


The theoretical contribution feels somewhat shallow. The connection in Section 4.1 is essentially saying "if you treat cached gradients as estimates of advantage gradients and use importance sampling, you get off-policy updates." This is almost a tautology - the interesting question would be characterizing when/why this actually helps, not just showing it's mathematically valid.

The clipping mechanism in Eq. (11) is introduced without any theoretical justification for why this particular form is appropriate for RPG. The paper says it's "tailored for RPG" but the asymmetric clipping with Clow and Chigh is essentially heuristic. Why not just use the PPO-style clipping directly? The paper says "RPG doesn't explicitly increase/decrease log-likelihood" but that's not a satisfying explanation.


The benchmark suite is limited - only 5 tasks from 2 simulators, all of which are fairly standard locomotion/manipulation tasks. Notably missing: tasks with stiffer contact dynamics, longer horizons, or environments where RPG is known to struggle. The paper acknowledges RPG has issues with "non-smooth dynamics" but doesn't test in such regimes.

The paper shows results with 12 seeds, running each twice for "24 effective runs" - but this phrasing is potentially misleading. Running the same seed twice accounts for simulator stochasticity but does not increase the number of independent statistical samples.

The paper doesn't discuss failure modes. Under what conditions does RPO fail? All the experiments show near-monotonic improvement - this is suspiciously clean.

While hyperparameter sensitivity is deferred to Appendix I, it would strengthen the main paper to include key results on how the number of reuse epochs M affects performance - when does sample reuse help vs. hurt?

---

> ### Author Rebuttal · Authors · 2026-03-29
>
> Thank you for your time and effort in reviewing our paper! We are grateful for your constructive suggestions, which have significantly guided our improvements.
>
> ---
>
> #### **Concern 1: On the Connection Between RPG and Surrogate Objectives**
>
> Regarding this concern, we respectfully argue that the connection is not shallow, but rather **beautiful and elegant**.
>
> The most intuitive way to understand RPO is through the **computational graph**. In standard RPG, gradients are backpropagated from the actions to policy parameters exactly once per rollout. Our work demonstrates a more fundamental operation: one can allow the gradients to flow from these action nodes to policy parameters **multiple times** for updates. This is a fundamental and principled operation on the computational graph that naturally links to the surrogate objective and sample reuse.
>
> #### **Concern 2: The Logic of Asymmetric Clipping**
>
> PPO's clipping mechanism depends on the sign of the advantage, as it "knows" whether an action should become more or less likely. However, RPG is significantly more **aggressive**. As noted in [1], even if the source noise $\epsilon$ is mapped to a "good" action, the gradient will still map the source noise to another action based on the gradient.
>
> Hence, it is not always clear whether an action's probability density should increase or decrease in this aggressive regime. To address this, we designed an **asymmetric clipping mechanism** which does not rely on the sign of the advantage. We will add more detailed explanations in the revised paper.
>
> #### **Concern 3, 5 & Q1: Non-smoothness and Failure Modes**
>
> To investigate performance under non-smooth dynamics, we increased the contact stiffness ($K_e$) on the **Hopper** task (default is 2e4). This directly controls the stiffness and non-smoothness of the contact dynamics.
>
> First, we tested at **$K_e = 5 \times 10^5$ (25x default)**, which represents a highly stiff regime. RPO maintains a clear advantage over SAPO and SHAC, demonstrating superior robustness to stiff contacts:
>
> | Method | Score (mean ± std) |
> | :--- | :--- |
> | **RPO (Ours)** | **4207.0 ± 1434.0** |
> | SAPO | 2120.9 ± 1809.1 |
> | SHAC | 2140.3 ± 2096.7 |
>
> **Failure Mode**: We further tested extreme stiffness ($K_e = 1 \times 10^6$). All RPG-based methods fail to produce meaningful results, as the underlying dynamics gradients become too ill-defined. This is a known limitation of the RPG paradigm, not specific to RPO, and is a valuable direction for future work.
>
>
> #### **Concern 4: Clarification**
> We apologize for the confusion. We will state the relationship between the components more clearly in the revised paper.
>
> #### **Concern 6: Limits of Sample Reuse**
>
> We conducted experiments on the **HandReorient** task to identify the limits of sample reuse. Performance begins to decline when the number of update epochs is increased to 10 or 15 (our default is **$M=5$**), suggesting that excessive updates lead to **gradient staleness** or over-optimization:
>
> | Method | Mean ± Std |
> | :--- | :--- |
> | **RPO (M=5)** | **237.5 ± 25.2** |
> | RPO (M=10) | 207.6 ± 30.1 |
> | RPO (M=15) | 175.1 ± 31.9 |
>
> #### **Q2: Robustness of Clipping Hyperparameters**
>
> We fixed **$c_{low}=0.8$** and **$c_{high}=1.0$** across all tasks. The consistent performance suggests they are robust and task-agnostic. Additionally, we provide ablations on these values for the **ANYmal** task in **Figure 11** of the Appendix, which further confirms RPO's stability across a range of clipping values.
>
> #### **Q3: Incorporating Results**
> Thank you for this constructive suggestion. We will follow your advice and incorporate these ablation results into the revised manuscript.
>
> ---
>
> **References:**
> [1] A unified view of likelihood ratio and reparameterization gradients, AISTATS 2021.
>
> We hope these clarifications address your concerns. If so, we wonder if you could kindly consider **raising your score**? We will also be happy to answer any further questions you may have. Thank you very much!

---

> > ### Author Rebuttal · Reviewer_9iea · 2026-04-04
> >
> > Overall, the experimental concerns have been addressed well.

---

> > > ### Author Response · Authors · 2026-04-04
> > >
> > > Thanks for your reply. Could you please let us know what are your remaining concerns? We hope we could further discuss it. Thanks so much for your time.

---

### Official Review · Reviewer_9FUS · 2026-03-12

**Soundness:** 3
**Presentation:** 3
**Significance:** 3
**Originality:** 2
**Overall Recommendation:** 4
**Confidence:** 4

**Summary:**

This paper proposes Reparameterization Proximal Policy Optimization (RPO), a PPO-style policy optimization method for reinforcement learning with differentiable dynamics. One limitation in previous works of RL with differentiable dynamics is that those methods rely on on-policy policy update and the dynamics Jacobian cannot be reused in an off-policy manner. This limitation results in low sample complexity in training and high computational cost especially for computing the Jacobian. To solve this issue, this work derives one reparameterization gradient formula with action-gradient reuse, introducing noise recovering and one importance sampling ratio to make the calculation tractable and unbiased. Experiments show the proposed method outperform classic online model-free RL algorithms such as PPO and RPG-based algorithms.

**Compliance With Llm Reviewing Policy:**

Affirmed.

**Final Justification:**

All my concerns have been adequately addressed in the rebuttal. I've raised the score from 3 to 4.

**Key Questions For Authors:**

1. Could you provide a full rigorous proof for the multidimensional case in Proposition D.1?
2. In Figure 4, why were the experiments of the ablation study on clipping not conducted on the Humanoid environment? Are there any specific reasons for this?
3. In Figure 4, after removing the sample reuse, why does the RPO_no_sample_reuse version still perform better than SHAC in Figure 3? Does this mean the claimed main contribution in the work does not contribute much to the empirical performance?

If the authors can address my above questions and concerns, I would be willing to raise the score.

**Limitations:**

The limitations of this work should be discussed more thoroughly.

**Strengths And Weaknesses:**

Strengths
1. This work is well-motivated. The backpropagation through time in reparameterization policy optimization induces large computational cost in training and the pure on-policy setting can reduce the sample efficiency. This paper aims to solve this limitation in this line of works.
2. The paper is overall well-written.

Weaknesses
1. This paper is a straightforward extension to previous papers in reparameterization policy optimization with differentiable dynamics.
2. The proof in Proposition D.1 mainly focuses on the one-dimensional case and does not provide a full proof for the multi-dimensional case.
3. It should be better explained in the main text why the function $f_\theta(a;s)$ is invertible, why there always exists a noise for corresponding to the action in the new policy, and how to obtain the recovered noise. These are due to the properties of the Gaussian model but should be explained more explicitly in the paper.

---

> ### Author Rebuttal · Authors · 2026-03-29
>
> Thank you for your time and effort in reviewing our paper! We are grateful for your constructive suggestions.
>
> ---
>
> #### **W1: Innovation and Significance**
>
> We respectfully argue that RPO is a **non-trivial advancement**, not a "straightforward extension":
>
> * **[i] Theoretical Bridge**: We establish a theoretical connection between PPO-style surrogate objectives and RPG, a link previously unexplored in the literature.
> * **[ii] Fixing Broken Graphs**: Unlike PPO, off-policy RPG faces a broken computational graph between old actions and new parameters. We manage to reconnect the computational graph and derive **Proposition 1** to ensure unbiased gradient estimation.
> * **[iii] Tailored Stability**: RPG gradients are more aggressive than REINFORCE. We designed a tailored system using **KL regularization and a custom clipping mechanism** specifically for RPG.
> * **[iv] Principled Framework**: Together, these elements constitute a principled framework for RPG gradient reuse, which significantly boosts sample efficiency to reach high performance scores.
>
> #### **W2 & Q1: Unified Proof**
>
> Thanks! While we initially presented the proofs separately, they can indeed be unified into a single, elegant proof for the n-dimensional case. As long as the reparameterization transformation is differentiable and bijective, the proof follows:
>
> 1.  **Density of Regenerated Noise**: We define the regenerated noise $\epsilon_{reg}$ via the mapping $\epsilon \to a \to \epsilon_{reg}$, where $a = f_{old}(\epsilon) = f_{new}(\epsilon_{reg})$. Let $J = \frac{\partial f}{\partial \epsilon}$. By the change of variables formula and the chain rule, the density $p_{reg}(\epsilon_{reg})$ is:
>     $$p_{reg}(\epsilon_{reg}) = p_{std}(\epsilon) \cdot \left| \det \left( \frac{\partial \epsilon}{\partial \epsilon_{reg}} \right) \right| = p_{std}(\epsilon) \cdot \frac{|\det J_{new}|}{|\det J_{old}|}$$
>
> 2.  **Symmetry in Importance Ratio**: The policy density $\pi(a)$ is defined as $p_{std}(\epsilon) \cdot |\det J|^{-1}$. Thus, the importance ratio $\rho(a)$ satisfies:
>     $$\rho(a) = \frac{\pi_{new}(a)}{\pi_{old}(a)} = \frac{p_{std}(\epsilon_{reg}) \cdot |\det J_{new}|^{-1}}{p_{std}(\epsilon) \cdot |\det J_{old}|^{-1}} = \frac{p_{std}(\epsilon_{reg})}{p_{std}(\epsilon)} \cdot \frac{|\det J_{old}|}{|\det J_{new}|}$$
>
> 3.  **The Symmetric Cancellation**: Multiplying these terms, the total effective density for the RPO update simplifies perfectly:
>     $$p_{reg}(\epsilon_{reg}) \cdot \rho(a) = \left( p_{std}(\epsilon) \cdot \frac{|\det J_{new}|}{|\det J_{old}|} \right) \cdot \left( \frac{p_{std}(\epsilon_{reg})}{p_{std}(\epsilon)} \cdot \frac{|\det J_{old}|}{|\det J_{new}|} \right) = p_{std}(\epsilon_{reg})$$
>
>
> #### **W3: Explanations**
> Thanks. Given $a = \mu_{\theta'}(s) + \sigma_{\theta'}(s) \cdot \epsilon_{reg}$, the noise is uniquely recovered as $\epsilon_{reg} = \sigma_{\theta'}(s)^{-1} \cdot (a - \mu_{\theta'}(s))$. Under our diagonal Gaussian policy, $\sigma_{\theta'}(s)$ is diagonal and always invertible since $\sigma_i > 0, \forall i$. We will make this explicit in the revised main text.
>
> #### **Q2: Clipping Ablation and Numerical Safeguards**
>
> We conduct an additional ablation study on the **Humanoid** task without the clipping mechanism:
> * **Instability**: In the early training phase, the maximum KL divergence reaches approximately **1.5** (compared to **~0.5** with clipping). While this slows down initial learning, it does not significantly degrade final converged performance in this environment.
> * **Stress Test**: We initially focused on **HandReorient** because it represents a much more rigorous "stress test" for numerical stability. Importance weight ratios in HandReorient can reach scales up to **~6000**, whereas they typically remain around **~600** in Humanoid. This demonstrates clipping's critical role as a numerical safeguard.
>
> #### **Q3: Impact of Sample Reuse vs. SHAC**
>
> The performance of `RPO_no_sample_reuse` over SHAC on the Humanoid task is due to the inclusion of entropy regularization in RPO, which is known to improve performance (as established in the SAPO paper).  And we compared with SAPO in our experiments and showed the value of our work.
>
> To further isolate the value of sample reuse, we conducted a new ablation on the **Hopper** task. Without sample reuse, RPO performs worse than SHAC, whereas with sample reuse, RPO outperforms it:
>
> | Method | Mean ± Std |
> | :--- | :--- |
> | **RPO (Ours)** | **5525.6 ± 3.5** |
> | RPO (No Sample Reuse) | 4803.0 ± 275.5 |
> | SHAC | 5067.2 ± 299.4 |
>
> With the sample reuse ablations on Hopper and Humanoid tasks, we clearly demonstrate that our proposed sample reuse mechanism is **critical** .
>
> ---
>
> We hope these clarifications address your concerns. If so, we wonder if you could kindly consider raising your score? We are happy to answer any further questions you may have.

---

> > ### Author Rebuttal · Reviewer_9FUS · 2026-04-04
> >
> > Thank you for the detailed reply. All my concerns have been adequately addressed and I have updated my score accordingly.

---

> > > ### Author Response · Authors · 2026-04-06
> > >
> > > Thanks again for your time and effort for reviewing our paper!

---

### Official Review · Reviewer_yGzU · 2026-03-12

**Soundness:** 3
**Presentation:** 3
**Significance:** 3
**Originality:** 3
**Overall Recommendation:** 5
**Confidence:** 3

**Summary:**

This paper considers a reparameterization of policy gradient (RGP) for model-based RL with differentiable simulators. The key idea of the paper is to compute policy gradient by backpropagating through the system dynamics (trajectories) instead of doing REINFORCE which computes gradients using the score function estimator.  Although computing gradients through the simulator would typically require expensive computation of the Jacobian, the paper demonstrates that reusing action-gradients in RPG is the same as optimizing a PPO objective, which allows for reusing samples and thus reduce sample complexity. The paper compares RPO on five continuous control tasks (Hopper,  Ant, Humanoid, Anymal, Hand Reorientation) using differentiable simulators, and compared to the baseline it provides higher sample efficiency and state of the art performance.

**Compliance With Llm Reviewing Policy:**

Affirmed.

**Key Questions For Authors:**

Comments: It would be nice to see how that approach works with learned world models as well, like differentiating over the trajectories of a learned world model for gradient based planning. In the same note, how does the approach would perform with a learned dynamics and compounding error in the trajectories? I would appreciate if the authors can elaborate some more on this.

Another question is in the experiments is there a clear limit for the number of reusing epochs before it degrades performance due to staleness?

**Limitations:**

yes

**Strengths And Weaknesses:**

Strength: The paper is very well-written and the sample efficiency improvement is relevant to the literature of model-based RL. The paper makes this clean theoretical connection between Reparameterization Policy Gradient (RPG) and PPO, which is not only theoretically interesting and relevant but also provide a principled way to reduce sample complexity and improve instability in RPG.

Weakness: The main concern I have with the applicability of the paper is that RPO only applies to differentiable simulators, however, in practice we typically have black-box environments that are not differentiable. This reduces the applicability of the approach.

---

> ### Author Rebuttal · Authors · 2026-03-29
>
> Thank you for your time and effort in reviewing our paper! We are grateful for your constructive suggestions, which have significantly guided our improvements.
>
> ---
>
> #### **W1 & Comment 1: On Differentiable Simulators and Learned Dynamics**
>
> First, we emphasize that training with differentiable simulators has already proven valuable for robotics applications, such as quadrotor flights [1,3] and quadrupedal locomotion [2,4,5]. Thus, we believe the current application of RPO within differentiable simulators provides significant practical value to the community.
>
> Second, we fully agree that integration with learnable dynamics models is of great value, and we thank you for this great suggestion! This represents a promising direction for future work. There is ongoing research into designing learned world models specifically tailored for RPG [6,7], which can bypass compounding errors by utilizing the learned model solely for the gradient computation in the backward pass. Furthermore, we believe that leveraging generative models for dynamics modeling is a highly significant direction for RPG. Flow matching models could possibly produce smooth dynamics gradients, which would be significant for RPG.
>
> ---
>
> #### **Q2: On the Limits of Sample Reuse**
>
> **Yes**, we observe a clear limit to sample reuse; exceeding this limit can degrade performance. To illustrate this, we conducted experiments on the **HandReorient** task. We found that performance begins to decline when the number of update epochs is increased to 10 or 15 (our default is $M=5$). This suggests that while sample reuse boosts efficiency, excessive updates may lead to gradient staleness or over-optimization on a single batch of samples, which can hurt the performance. The final performance results are summarized in the table below:
>
> | Method | Mean ± Std |
> | :--- | :--- |
> | **RPO (M=5)** | **237.5 ± 25.2** |
> | RPO (M=10) | 207.6 ± 30.1 |
> | RPO (M=15) | 175.1 ± 31.9 |
>
> ---
>
> **References:**
> [1] Learning on the fly: Rapid policy adaptation via differentiable simulation
>
>
> [2] Learning quadruped locomotion using differentiable simulation
>
>
> [3] Back to Newton’s laws: Learning vision-based agile flight via differentiable physics
>
>
> [4] Residual policy learning for perceptive quadruped control using differentiable simulation
>
>
> [5] Diffsim2real: Deploying quadrupedal locomotion policies purely trained in differentiable simulation
>
>
> [6] First order model-based RL through decoupled backpropagation
>
> [7] Coupled Local and Global World Models for Efficient First Order RL
>
> Thank you very much! We hope these clarifications address your concerns. If so, we wonder if you could kindly consider **raising your confidence**?

---

> > ### Author Rebuttal · Reviewer_yGzU · 2026-04-03
> >
> > I thank the authors for addressing all my concerns.

---

> > > ### Author Response · Authors · 2026-04-06
> > >
> > > Thanks again for your time and effort for reviewing our paper!

---

### Official Review · Reviewer_buvU · 2026-03-17

**Soundness:** 3
**Presentation:** 3
**Significance:** 3
**Originality:** 3
**Overall Recommendation:** 4
**Confidence:** 3

**Summary:**

This paper introduces Reparameterization Proximal Policy Optimization (RPO), that reuse past samples without reducing stability. The main idea is pretty straightforward. The key is to reuse action gradients computed by backpropagation through time. This gives a way to connect on-policy and off-policy updates in RPG. Since RPG is known to be unstable, the authors introduce an importance-weighted ratio clipping mechanism and add KL regularization. In experiments on several continuous-control tasks using DFlex and Rewarped, RPO shows consistently more stable training and also achieves high efficiency on data usage. It also reaches better final performance than the model-free baselines.

**Compliance With Llm Reviewing Policy:**

Affirmed.

**Ethical Review Concerns:**

Please refer to the weaknesses

**Key Questions For Authors:**

Weaknesses:


The RPO introduces task-specific hyperparameters, such as the entropy coefficient, the KL coefficient, and the asymmetric clipping bounds $c_{\text{low}}$ and $c_{\text{high}}$, which depend on the particular task. The author didn't discuss how to select them for a new task.


The reported PPO baseline on Humanoid is 1665, which appears lower than what is typically seen in standard benchmarks. It would be helpful if the authors could clarify the reasons behind this.


The theoretical assumptions assume deterministic dynamics and differentiable rewards. These are strong requirements for the environments. The author uses the DFlex to make smooth contact approximations.  This post raises a question about whether the RPO stability still holds with contact discontinuities.


The paper also misses some closely related baselines, such as the Adaptive-Gradient Policy Optimization, AHAC.

**Limitations:**

Please refer to the weaknesses

**Strengths And Weaknesses:**

Strengths:

The reuse of cached action gradients with importance weighting provides a computationally efficient way to address the expensive Jacobian computations across multiple policy updates.

The clipping strategy is designed for RPG. Instead of depending on advantage signs like PPO, it focuses on controlling the importance ratio. This combined with the KL regularization, it directly addresses the issues for RPG.

The experimental setup is solid. The authors compare against strong baselines, run 12 seeds per task, and include ablation studies that clearly show the impact of sample reuse, clipping, and KL regularization.

The paper is also clearly written and easy to follow. The algorithm is fully specified in pseudocode, and details about the training pipeline, wall-clock time, and hardware are all included, making it much easier to reproduce the results.

---

> ### Author Rebuttal · Authors · 2026-03-29
>
> Thank you for your time and effort in reviewing our paper! We are grateful for your constructive suggestions, which have significantly guided our improvements.
>
> ---
>
> #### **W1: Hyperparameters**
>
> **Clipping:** The role of clipping is to filter out samples with large importance weight ratios to avoid numerical instability, and to prevent the probability of certain actions from becoming too low. We fixed $c_{low}=0.8$ and $c_{high}=1.0$ across all tasks. The consistent performance suggests that $c_{low}=0.8$ and $c_{high}=1.0$ are robust, task-agnostic and suitable for new tasks.
>
> **KL and Entropy Coefficients:** While task-dependent, $\alpha$ and $\beta$ operate on the same scale. For new tasks, an effective strategy is to initialize both with the same value and then perform tuning. This heuristic is reflected in the values reported in Table 4. We also have ablations on hyperparameters in Appendix Section I.
>
> ---
>
> #### **W2: PPO**
>
> **Implementation & Consistency:** We used the PPO implementation from the SAPO codebase with aligned MLP sizes. Our results are consistent with those reported in the original SAPO paper.
>
> **Sample Efficiency:** It is known that PPO is less sample-efficient than RPG methods. To address this concern, we trained PPO for **100M** environment steps on the **Humanoid** task, and it achieved a score of **5100**, which remains lower than RPO with only **10M** steps. This highlights the fundamental sample efficiency advantage of RPG-based approaches.
>
> **Focus:** Our primary contribution is advancing RPG-based methods, where RPO is compared against SOTA RPG baselines (SAPO, SHAC, GIPPO). We include PPO as a reference baseline to contextualize the sample efficiency gap between the two paradigms.
>
> ---
>
> #### **W3: Non-smooth Dynamics**
>
> Thanks for this great question. To investigate performance under non-smooth dynamics, we increased the contact stiffness ($K_e$) on the Hopper task (default is 2e4). This directly controls the stiffness and non-smoothness of the contact dynamics.
>
> At $K_e = 5 \times 10^5$ (25x default), which represents a highly stiff regime, RPO maintains a clear advantage over SAPO and SHAC, demonstrating superior robustness to stiff contacts:
>
> | Method | Score (mean ± std) |
> | :--- | :--- |
> | **RPO (Ours)** | **4207.0 ± 1434.0** |
> | SAPO | 2120.9 ± 1809.1 |
> | SHAC | 2140.3 ± 2096.7 |
>
> We further tested extreme stiffness ($K_e = 1 \times 10^6$). All RPG-based methods fail to produce meaningful results, as the underlying dynamics gradients become too ill-defined. This is a known limitation of the RPG paradigm, not specific to RPO, and is a valuable direction for future work.
>
> ---
>
> #### **W4: AHAC and AGPO**
>
> **AHAC:** We evaluated AHAC using its official repository and hyperparameters on the Anymal task with 6 random seeds. We found that the adaptive horizon $H$ converged to either 8 or 64. Specifically, **5 seeds converged to $H=64$**, where training exhibited significant instability due to gradient explosion, producing NaN issues that led to quick early termination. For the remaining seed that converged to $H=8$, the performance was suboptimal, consistent with results previously shown in the SHAC paper.
>
> **AGPO:** To the best of our knowledge, AGPO is not open-source. We attempted to incorporate its Q-gradient logic into our framework but observed no performance improvement in our test cases.
>
> **Complementarity:** RPO’s contributions are orthogonal to horizon adaptation (AHAC) or the use of Q-gradients (AGPO).  RPO is compatible with these techniques. Our current comparisons and ablations already clearly demonstrate the unique value and contributions of RPO.
>
> ---
>
> We hope these clarifications address your concerns. If so, we wonder if you could kindly consider raising your score? We are happy to answer any further questions you may have. Thank you very much!

---

> > ### Author Rebuttal · Reviewer_buvU · 2026-04-04
> >
> > Thank you for the detailed rebuttal.
> >
> > The rebuttal for W1 claims that KL and entropy coefficients operate on the same scale, but Table 4 shows the entropy coefficient ranging from 0.001 to 0.5 across tasks.
> >
> > The value function is only updated after all M policy epochs finish, meaning advantage estimates in later epochs are computed under an increasingly outdated value function. Does this staleness affect gradient quality, and would the author consider more frequent value-function updates?

---

> > > ### Author Response · Authors · 2026-04-04
> > >
> > > Thanks for the follow-up questions!
> > >
> > > ## 1: Entropy and KL Coefficient Scale
> > >
> > > We apologize for the confusion and should have stated this more clearly. We mean that *for the same task*, the two coefficients (entropy and KL) are on the same scale, which is consistent with the hyperparameter table (Table 4).
> > >
> > > Additionally, the Hand Reorientation task differs from the other tasks in that its cumulative reward is roughly within 0–300, significantly lower than other tasks. Hence, the KL and entropy coefficients must be much smaller, so that their gradients do not dominate the 32 horizon's SHAC-style BPTT gradients (We also helped tune the initial entropy coefficient for SAPO on this task, as it was too high in their official implementation and paper when they conduct this handreorient experiment).
> > >
> > > ## 2: Value Function Staleness During Policy Epochs
> > >
> > > Thanks for this insightful question! We address it from several angles below.
> > >
> > > **Clarification on the objective.** First, we clarify that our main policy optimization objective is:
> > >
> > > $$
> > > L\_{\pi\_{\theta\_{\text{old}}}}(\theta) = \mathbb{E}\_{s \sim d^{\pi\_{\theta\_{\text{old}}}},\, \varepsilon \sim p\_{\text{std}}} \left[ A^{\pi\_{\theta\_{\text{old}}}}(s,\, f\_{\theta}(\varepsilon;\, s)) \right]
> > > $$
> > >
> > > which, by definition, measures performance under the *behavior policy* used to collect samples. Although this is a surrogate objective, as long as the optimized policy remains close to the behavior policy, optimizing this surrogate effectively optimizes the true return, this is precisely the role of our KL regularization term [1].
> > >
> > > **i. No explicit advantage calculation.** Second, unlike standard PPO, we do not need to explicitly compute the advantage $A^{\pi}(s, a)$ itself. Instead, we compute **gradients of the advantage with respect to actions**, which, as shown in our paper, can be obtained via backpropagation through time (BPTT) in the SHAC-style to yield action-gradients.
> > >
> > > **ii. Role of the critic.** Third, the critic estimate appears as a *bootstrap* appended to the truncated return in the SHAC-style rollout. With our 32-step horizon, the critic provides $V(s_{H})$ to estimate future returns beyond the horizon. Hence, the "staleness" concern applies specifically to this bootstrap term within the action-gradient.
> > >
> > >
> > > **iii. Hence, if one updates the critic alongside policy optimization, it is not optimizing this surrogate objective, and it is not our proposed method.**
> > >
> > > ### Attempt: Refreshing the Bootstrap with an Updated Critic
> > >
> > > Following your suggestion, we investigated whether updating the critic *during* the policy optimization phase and recomputing the bootstrap gradient could improve gradient quality. The idea is to interleave critic updates within the policy update epochs, and then use the updated critic to re-estimate the bootstrap value $V(s_H)$, so that the action-gradient reflects a fresher value function.
> > >
> > > We tried our best in this limited amount of time. However, this turned out to be non-trivial in practice. Because the action-gradients are obtained via BPTT through the differentiable simulation, the entire trajectory's computational graph must remain available for refreshed critic's estimations gradient backwardpass. The computational graph for BPTT must be retained across optimization steps for later gradient recomputation. Such long-lived graphs are fragile; in-place tensor operations that commonly occur during gradient-based optimization (e.g., parameter updates, gradient accumulation) can invalidate the retained graph at runtime.
> > >
> > > ### Simpler Alternative: Increasing Critic Update Iterations
> > >
> > > As a more direct test, we also experimented with simply increasing the number of critic update iterations per epoch (from the default 16 to 32). The results on the Hopper task show no improvement:
> > >
> > > | Critic Iterations | Episode Reward |
> > > |---|---|
> > > | 16 (default) | 5525.57 ± 3.47 |
> > > | 32 | 5534.32 ± 223.68 |
> > >
> > > This suggests that the default critic training is already sufficient to provide a good bootstrap estimate.
> > >
> > > [1] Kakade, S. and Langford, J. *Approximately Optimal Approximate Reinforcement Learning.* ICML, 2002.
> > >
> > > -----
> > > We hope these clarifications address your concerns. If so, we wonder if you could kindly consider raising your score? We are happy to answer any further questions you may have. Thank you very much!

---

### Decision · Program_Chairs · 2026-04-30

**Decision:**

Accept (regular)

**Comment:**

The paper advances a specific sub-area (RL with differentiable simulators) with a principled and practically useful contribution. The theoretical connection is elegant, and the method demonstrates clear empirical gains.

The paper is likely to be built upon by researchers working on simulator-based RL and differentiable physics.

Thus i recommend the paper for acceptance (but with some descretion).

The paper can be still improved in:
- The Hyperparameter sensitivity / task‑dependence is commonly shared by rev buvU and 9iea;
- Rev 9iea also has a concern about shallow theoretical contribution (“connection is almost a tautology”);
- concerns not addressed by the rebuttal are value function staleness (by rev buvU) and (clipping mechanism is heuristic; by rev 9iea).

Hope these can be addressed in the future versions.